# Bayesian Optimization under Heavy-tailed Payoffs

**Sayak Ray Chowdhury**
Department of ECE
Indian Institute of Science
Bangalore, India 560012
sayak@iisc.ac.in

**Aditya Gopalan**
Department of ECE
Indian Institute of Science
Bangalore, India 560012
aditya@iisc.ac.in

## Abstract

We consider black box optimization of an unknown function in the nonparametric Gaussian process setting when the noise in the observed function values can be heavy tailed. This is in contrast to existing literature that typically assumes sub-Gaussian noise distributions for queries. Under the assumption that the unknown function belongs to the Reproducing Kernel Hilbert Space (RKHS) induced by a kernel, we first show that an adaptation of the well-known GP-UCB algorithm with reward truncation enjoys sublinear $\tilde{O}\left(T^{\frac{2+\alpha}{2(1+\alpha)}}\right)$ regret even with only the $(1 + \alpha)$-th moments, $\alpha \in (0, 1]$, of the reward distribution being bounded ($\tilde{O}$ hides logarithmic factors). However, for the common squared exponential (SE) and Matérn kernels, this is seen to be significantly larger than a fundamental $\Omega(T^{\frac{1}{1+\alpha}})$ lower bound on regret. We resolve this gap by developing novel Bayesian optimization algorithms, based on kernel approximation techniques, with regret bounds matching the lower bound in order for the SE kernel. We numerically benchmark the algorithms on environments based on both synthetic models and real-world data sets.

## 1 Introduction

Black-box optimization of an unknown function $f : \mathbb{R}^d \to \mathbb{R}$ with expensive, noisy queries is a generic problem arising in domains such as hyper-parameter tuning for complex machine learning models [3], sensor selection [14], synthetic gene design [15], experimental design etc. The popular Bayesian optimization (BO) approach, towards solving this problem, starts with a prior distribution, typically a nonparametric Gaussian process (GP), over a function class, uses function evaluations to compute the posterior distribution over functions, and chooses the next function evaluation adaptively – using a sampling strategy – towards reaching the optimum. Popular sampling strategies include expected improvement [25], probability of improvement [40], upper confidence bounds [35], Thompson sampling [11], predictive-entropy search [17], etc.

The design and analysis of adaptive sampling strategies for BO typically involves the assumption of bounded, or at worst sub-Gaussian, distributions for rewards (or losses) observed by the learner, which is quite light-tailed. Yet, many real-world environments are known to exhibit heavy-tailed behavior, e.g., the distribution of delays in data networks is inherently heavy-tailed especially with highly variable or bursty traffic flow distributions that are well-modeled with heavy tails [20], heavy-tailed price fluctuations are common in finance and insurance data [29], properties of complex networks often exhibit heavy tails such as degree distribution [37], etc. This motivates studying methods for Bayesian optimization when observations are significantly heavy tailed compared to Gaussian.

A simple version of black box optimization – in the form of online learning in finite multi-armed bandits (MABs) – with heavy-tailed payoffs, was first studied rigorously by Bubeck et al. [8], where the payoffs are assumed to have bounded $(1 + \alpha)$-th moment for $\alpha \in (0, 1]$. They showed that for

MABs with only finite variances (i.e., $\alpha = 1$), by using statistical estimators that are more robust than the empirical mean, one can still recover the optimal regret rate for MAB under the sub-Gaussian assumption. Moving further, Medina and Yang [24] consider these estimators for the problem of linear (parametric) stochastic bandits under heavy-tailed rewards and Shao et al. [34] show that almost optimal algorithms can be designed by using an optimistic, data-adaptive truncation of rewards. Some other important works include pure exploration under heavy-tailed noise [43], payoffs with bounded kurtosis [23], extreme bandits [10], heavy tailed payoffs with $\alpha \in (0, \infty)$ [38].

Against this backdrop, we consider regret minimization with heavy-tailed reward distributions in bandits with a potentially continuous arm set, and whose (unknown) expected reward function is nonparametric and assumed to have smoothness compatible with a kernel on the arm set. Here, it is unclear if existing BO techniques relying on statistical confidence sets based on sub-Gaussian observations can be made to work to attain nontrivial regret, since it is unlikely that these confidence sets will at all be correct. It is worth mentioning that in the finite dimensional setting, Shao et al. [34] solve the problem almost optimally, but their results do not carry over to the general nonparametric kernelized setup since their algorithms and regret bounds depend crucially on the finite feature dimension. We answer this affirmatively in this work, and formalize and solve BO under heavy tailed noise almost optimally. Specifically, this paper makes the following contributions.

- We adapt the GP-UCB algorithm to heavy-tailed payoffs by a truncation step, and show that it enjoys a regret bound of $\tilde{O}(\gamma_T T^{\frac{2+\alpha}{2(1+\alpha)}})$ where $\gamma_T$ depends on the kernel associated with the RKHS and is generally sub-linear in $T$. This regret rate, however, is potentially sub-optimal due to a $\Omega(T^{\frac{1}{1+\alpha}})$ fundamental lower bound on regret that we show for two specific kernels, namely the squared exponential (SE) kernel and the Matérn kernel.

- We develop a new Bayesian optimization algorithm by truncating rewards in each direction of an approximate, finite-dimensional feature space. We show that the feature approximation can be carried out by two popular kernel approximation techniques: Quadrature Fourier features [26] and Nyström approximation [9]. The new algorithm under either approximation scheme gets regret $\tilde{O}(\gamma_T T^{\frac{1}{1+\alpha}})$, which is optimal upto log factors for the SE kernel.

- Finally, we report numerical results based on experiments on synthetic as well as real-world based datasets, for which the algorithms we develop are seen to perform favorably in the harsher heavy-tailed environments.

**Related work.** An alternative line of work uses approaches for black box optimization based on Lipschitz-type smoothness structure [22, 7, 2, 33], which is qualitatively different from RKHS smoothness type assumptions. Recently, Bogunovic et al. [5] consider GP optimization under an adversarial perturbation of the query points. But, the observation noise is assumed to be Gaussian unlike our heavy-tailed environments. Kernel approximation schemes in the context of BO usually focuses on reducing the cubic cost of gram matrix inversion [39, 41, 26, 9]. However, we crucially use these approximations to achieve optimal regret for BO under heavy tailed noise, which, we believe, might not be possible without resorting to the kernel approximations.

## 2 Problem formulation

Let $f : \mathcal{X} \to \mathbb{R}$ be a fixed but unknown function over a domain $\mathcal{X} \subset \mathbb{R}^d$ for some $d \in \mathbb{N}$. At every round, a learner queries $f$ at a single point $x_t \in \mathcal{X}$, and observes a noisy payoff $y_t = f(x_t) + \eta_t$. Here the noise sequence $\eta_t, t \geq 1$ are assumed to be zero mean i.i.d. random variables such that the payoffs satisfy $\mathbb{E}\left[|y_t|^{1+\alpha} | \mathcal{F}_{t-1}\right] \leq v$ for some $\alpha \in (0, 1]$ and $v \in (0, \infty)$, where $\mathcal{F}_{t-1} = \sigma(\{x_\tau, y_\tau\}_{\tau=1}^{t-1}, x_t)$ denotes the $\sigma$-algebra generated by the events so far[1]. Observe that this bound on the $(1 + \alpha)$-th moment at best yields bounded variance for $y_t$, and does not necessarily mean that $y_t$ (or $\eta_t$) is sub-Gaussian as is assumed typically. The query point $x_t$ at round $t$ is chosen causally depending upon the history $\{(x_s, y_s)\}_{s=1}^{t-1}$ of query and payoff sequences available up to round $t - 1$. The learner's goal is to maximize its (expected) cumulative reward $\sum_{t=1}^{T} f(x_t)$ over a time horizon $T$ or equivalently minimize its cumulative *regret* $R_T = \sum_{t=1}^{T} (f(x^\star) - f(x_t))$, where

$x^\star \in \mathrm{argmax}_{x \in \mathcal{X}} f(x)$ is a maximum point of $f$ (assuming the maximum is attained; not necessarily unique). A sublinear growth of $R_T$ with $T$ implies the time-average regret $R_T/T \to 0$ as $T \to \infty$.

**Regularity assumptions:** Attaining sub-linear regret is impossible in general for arbitrary reward functions $f$, and thus some regularity assumptions are needed. In this paper, we assume smoothness for $f$ induced by the structure of a kernel on $\mathcal{X}$. Specifically, we make the standard assumption of a p.s.d. kernel $k : \mathcal{X} \times \mathcal{X} \to \mathbb{R}$ such that $k(x, x) \leq 1$ for all $x \in \mathcal{X}$, and $f$ being an element of the reproducing kernel Hilbert space (RKHS) $\mathcal{H}_k(\mathcal{X})$ of smooth real valued functions on $\mathcal{X}$. Moreover, the RKHS norm of $f$ is assumed to be bounded, i.e., $\|f\|_{\mathcal{H}} \leq B$ for some $B < \infty$. Boundedness of $k$ along the diagonal holds for any stationary kernel, i.e., where $k(x, x') = k(x - x')$, e.g., the *Squared Exponential* kernel $k_{\mathrm{SE}}$ and the *Matérn* kernel $k_{\mathrm{Matérn}}$:

$$k_{\mathrm{SE}}(x, x') = \exp\left(-\frac{r^2}{2l^2}\right) \quad \text{and} \quad k_{\mathrm{Matérn}}(x, x') = \frac{2^{1-\nu}}{\Gamma(\nu)}\left(\frac{r\sqrt{2\nu}}{l}\right)^\nu B_\nu\left(\frac{r\sqrt{2\nu}}{l}\right),$$

where $l > 0$ and $\nu > 0$ are hyperparameters of the kernels, $r = \|x - x'\|_2$ is the distance between $x$ and $x'$, and $B_\nu$ is the modified Bessel function.

## 3 Warm-up: the first algorithm

Towards designing a BO algorithm for heavy tailed observations, we briefly recall the standard GP-UCB algorithm for the sub-Gaussian setting. GP-UCB at time $t$ chooses the point $x_t = \mathrm{argmax}_{x \in \mathcal{X}} \mu_{t-1}(x) + \beta_t \sigma_{t-1}(x)$ where $\mu_t(x) = k_t(x)^T (K_t + \lambda I_t)^{-1} Y_t$ and $\sigma_t^2(x) = k(x, x) - k_t(x)^T (K_t + \lambda I_t)^{-1} k_t(x)$ are the posterior mean and variance functions after $t$ observations from a function drawn from the GP prior $GP_{\mathcal{X}}(0, k)$, with additive i.i.d. Gaussian noise $\mathcal{N}(0, \lambda)$. Here $Y_t = [y_1, \ldots, y_t]^T$ is the vector formed by observations, $K_t = [k(u, v)]_{u,v \in \mathcal{X}_t}$ is the kernel matrix computed at the set $\mathcal{X}_t = \{x_1, \ldots, x_t\}$, $k_t(x) = [k(x_1, x), \ldots, k(x_t, x)]^T$ and $I_t$ is the identity matrix of order $t$. If the noise $\eta_t$ is assumed conditionally $R$-sub-Gaussian, i.e., $\mathbb{E}\left[e^{\gamma \eta_t} \mid \mathcal{F}_{t-1}\right] \leq \exp\left(\frac{\gamma^2 R^2}{2}\right)$ for all $\gamma \in \mathbb{R}$, then using $\beta_{t+1} = O\left(R\sqrt{\ln|I_t + \lambda^{-1} K_t|}\right)$ ensures $\tilde{O}(\sqrt{T})$ regret [11], as the posterior GP concentrates rapidly on the true function $f$. However, when the sub-Gaussian assumption does not hold, we cannot expect the posterior GP to have such nice concentration property. In fact, it is known that the ridge regression estimator $\mu_t \in \mathcal{H}_k(\mathcal{X})$ of $f$ is not robust when the noise exhibits heavy fluctuations [19]. So, in order to tackle heavy tailed noise, one needs more robust estimates $\widehat{\mu}_t$ of $f$ along with suitable confidence sets. A natural idea to curb the effects of heavy fluctuations is to truncate high rewards [8]. Our first algorithm Truncated GP-UCB (Algorithm 1) is based on this idea.

**Truncated GP-UCB (TGP-UCB) algorithm:**

At each time $t$, we truncate the reward $y_t$ to zero if it is larger than a suitably chosen truncation level $b_t$, i.e., we set the truncated reward $\widehat{y}_t = y_t \mathbb{1}_{|y_t| \leq b_t}$. Then, we construct the truncated version of the posterior mean as $\widehat{\mu}_t(x) = k_t(x)^T (K_t + \lambda I_t)^{-1} \widehat{Y}_t$ where $\widehat{Y}_t = [\widehat{y}_1, \ldots, \widehat{y}_t]^T$ and simply run GP-UCB with $\widehat{\mu}_t$ instead of $\mu_t$. The truncation level $b_t$ can be adapted with time $t$. We choose an increasing sequence of $b_t$'s, i.e., as time progresses and confidence interval shrinks, we truncate less and less

---

**Algorithm 1** Truncated GP-UCB (TGP-UCB)

**Input:** Parameters $\lambda > 0$, $\{b_t\}_{t \geq 1}$, $\{\beta_t\}_{t \geq 1}$
Set $\widehat{\mu}_0(x) = 0$ and $\sigma_0^2(x) = k(x, x) \forall x \in \mathcal{X}$
**for** $t = 1, 2, 3 \ldots$ **do**
  Play $x_t = \mathrm{argmax}_{x \in \mathcal{X}} \widehat{\mu}_{t-1}(x) + \beta_t \sigma_{t-1}(x)$
  and observe payoff $y_t$
  Set $\widehat{y}_t = y_t \mathbb{1}_{|y_t| \leq b_t}$ and $\widehat{Y}_t = [\widehat{y}_1, \ldots, \widehat{y}_t]^T$
  Compute $\widehat{\mu}_t(x) = k_t(x)^T (K_t + \lambda I_t)^{-1} \widehat{Y}_t$
  and $\sigma_t^2(x) = k_t(x)^T (K_t + \lambda I_t)^{-1} k_t(x)$
**end for**

---

aggressively. Finally, in order to account for the bias introduced by truncation, we blow up the confidence width $\beta_t$ of GP-UCB by a multiplicative factor of $b_t$ so that $f(x)$ is contained in the interval $\widehat{\mu}_{t-1}(x) \pm \beta_t \sigma_{t-1}(x)$ with high probability. This helps us to obtain a sub-linear regret bound for TGP-UCB given in the Theorem 1, with a full proof deferred to appendix B.

**Theorem 1 (Regret bound for TGP-UCB)** *Let* $f \in \mathcal{H}_k(\mathcal{X})$, $\|f\|_{\mathcal{H}} \leq B$ *and* $k(x, x) \leq 1$ *for all* $x \in \mathcal{X}$. *Let* $\mathbb{E}\left[|y_t|^{1+\alpha} \mid \mathcal{F}_{t-1}\right] \leq v < \infty$ *for some* $\alpha \in (0, 1]$ *and for all* $t \geq 1$. *Then, for any* $\delta \in (0, 1]$, *TGP-UCB, with* $b_t = v^{\frac{1}{1+\alpha}} t^{\frac{1}{2(1+\alpha)}}$ *and* $\beta_{t+1} = B + \frac{3}{\sqrt{\lambda}} b_t \sqrt{\ln|I_t + \lambda^{-1} K_t| + 2\ln(1/\delta)}$,

*enjoys, with probability at least $1 - \delta$, the regret bound*

$$R_T = O\left(B\sqrt{T\gamma_T} + v^{\frac{1}{1+\alpha}}\sqrt{\gamma_T\left(\gamma_T + \ln(1/\delta)\right)}T^{\frac{2+\alpha}{2(1+\alpha)}}\right),$$

*where $\gamma_T \equiv \gamma_T(k, \mathcal{X}) = \max_{A \subset \mathcal{X}:|A|=T} \frac{1}{2}\ln\left|I_t + \lambda^{-1}K_A\right|$.*

Here, $\gamma_T$ denotes the *maximum information gain* about any $f \sim GP_{\mathcal{X}}(0, k)$ after $T$ noisy observations obtained by passing $f$ through an i.i.d. Gaussian channel $\mathcal{N}(0, \lambda)$, and measures the reduction in the uncertainty of $f$ after $T$ noisy observations. It is a property of the kernel $k$ and domain $\mathcal{X}$, e.g., if $\mathcal{X}$ is compact and convex, then $\gamma_T = O\left((\ln T)^{d+1}\right)$ for $k_{SE}$ and $O\left(T^{\frac{d(d+1)}{2\nu+d(d+1)}}\ln T\right)$ for $k_{\text{Matérn}}$ [35].

**Remark 1.** An $R$-sub-Gaussian environment satisfies the moment condition with $\alpha = 1$ and $v = R^2$, so the result implies a sub-linear $\tilde{O}(T^{3/4})$ regret bound for TGP-UCB in sub-Gaussian environments.

## 4   Regret lower bound

Establishing lower bounds under general kernel smoothness structure is an open problem even when the payoffs are Gaussian. Similar to Scarlett et al. [31], we only focus on the SE and Matérn kernels.

**Theorem 2 (Lower bound on cumulative regret)** *Let $\mathcal{X} = [0,1]^d$ for some $d \in \mathbb{N}$. Fix a kernel $k \in \{k_{SE}, k_{Matérn}\}$, $B > 0$, $T \in \mathbb{N}$, $\alpha \in (0,1]$ and $v > 0$. Given any algorithm, there exists a function $f \in \mathcal{H}_k(\mathcal{X})$ with $\|f\|_{\mathcal{H}} \leq B$, and a reward distribution satisfying $\mathbb{E}\left[|y_t|^{1+\alpha}|\mathcal{F}_{t-1}\right] \leq v$ for all $t \in [T] := \{1, 2, \ldots, T\}$, such that when the algorithm is run with this $f$ and reward distribution, its regret satisfies*

*1.* $\mathbb{E}[R_T] = \Omega\left(v^{\frac{1}{1+\alpha}}\left(\ln\left(v^{-\frac{1}{\alpha}}B^{\frac{1+\alpha}{\alpha}}T\right)\right)^{\frac{d\alpha}{2(1+\alpha)}}T^{\frac{1}{1+\alpha}}\right)$ *if $k = k_{SE}$,*

*2.* $\mathbb{E}[R_T] = \Omega\left(v^{\frac{\nu}{\nu(1+\alpha)+d\alpha}}B^{\frac{d\alpha}{\nu(1+\alpha)+d\alpha}}T^{\frac{\nu+d\alpha}{\nu(1+\alpha)+d\alpha}}\right)$ *if $k = k_{Matérn}$.*

The proof argument is inspired by that of Scarlett et al. [31], which provides the lower bound of BO under i.i.d. Gaussian noise, but with nontrivial changes to account for heavy tailed observations. The proof is based on constructing a finite subset of "difficult" functions in $\mathcal{H}_k(\mathcal{X})$. Specifically, we choose $f$ as a uniformly sampled function from a finite set $\{f_1, \ldots, f_M\}$, where each $f_j$ is obtained by shifting a common function $g \in \mathcal{H}_k(\mathbb{R}^d)$ by a different amount such that each of these has a unique maximum, and then cropping to $\mathcal{X} = [0,1]^d$. $g$ takes values in $[-2\Delta, 2\Delta]$ with the maximum attained at $x = 0$. The function $g$ is constructed properly, and the parameters $\Delta$, $M$ are chosen appropriately based on the kernel $k$, fixed constants $B, T, \alpha, v$ such that any $\Delta$-optimal point for $f_j$ fails to be $\Delta$-optimal point for any other $f_{j'}$ and that $\|f_j\|_{\mathcal{H}} \leq B$ for all $j \in [M]$. The reward function takes values in $\{sgn\left(f(x)\right)\left(\frac{v}{2\Delta}\right)^{\frac{1}{\alpha}}, 0\}$, with the former occurring with probability $\left(\frac{2\Delta}{v}\right)^{\frac{1}{\alpha}}|f(x)|$, such that, for every $x \in \mathcal{X}$, the expected reward is $f(x)$ and $(1+\alpha)$-th raw moment is upper bounded by $v$. Now, if we can lower bound the regret averaged over $j \in [M]$, then there must exist some $f_j$ for which the bound holds. The formal proof is deferred to Appendix C.

**Remark 2.** Theorem 2 suggests that (a) TGP-UCB may be suboptimal, and (b) for the SE kernel, it may be possible to design algorithms recovering $\tilde{O}(\sqrt{T})$ regret bound under finite variances ($\alpha = 1$).

## 5   An optimal algorithm under heavy tailed rewards

In view of the gap between the regret bound for TGP-UCB and the fundamental lower bound, it is possible that TGP-UCB (Algorithm 1) does not completely mitigate the effect of heavy-tailed fluctuations, and perhaps that truncation in a different domain may work better. In fact, for parametric linear bandits (i.e., BO with finite dimensional linear kernels), it has been shown that appropriate truncation in feature space improves regret performance as opposed to truncating raw observations [34], and in this case the feature dimension explicitly appears in the regret bound. However, the main challenge in the more general nonparametric setting is that the feature space is infinite dimensional,

which would yield a trivial regret upper bound. If we can find an approximate feature map $\tilde{\varphi} : \mathcal{X} \to \mathbb{R}^m$ in a low-dimensional Euclidean inner product space $\mathbb{R}^m$ such that $k(x, y) \approx \tilde{\varphi}(x)^T \tilde{\varphi}(y)$, then we can perform the above feature adaptive truncation effectively as well as keep the error introduced due to approximation in control. Such a kernel approximation can be done efficiently either in a data independent way (Fourier features approximation [28]) or in a data dependent way (Nyström approximation [12]) and has been used in the context of BO to reduce the time complexity of GP-UCB [26, 9]. But in this work, the approximations are crucial to obtain optimal theoretical guarantees. We now describe our algorithm Adaptively Truncated Approximate GP-UCB (Algorithm 2).

## 5.1 Adaptively Truncated Approximate GP-UCB (ATA-GP-UCB) algorithm

At each round $t$, we select an arm $x_t$ which maximizes the approximate (under kernel approximation) GP-UCB score $\tilde{\mu}_{t-1}(x) + \beta_t \tilde{\sigma}_{t-1}(x)$, where $\tilde{\mu}_{t-1}(x)$ and $\tilde{\sigma}_{t-1}^2(x)$ denote approximate posterior mean and variance from the previous round, respectively and $\beta_t$ is an appropriately chosen confidence width. Then, we update $\tilde{\mu}_t(x)$ and $\tilde{\sigma}_t^2(x)$ as follows. First, we find a feature embedding $\tilde{\varphi}_t \in \mathbb{R}^{m_t}$, of some appropriate dimension $m_t$, which approximates the kernel efficiently. Then, we find the rows $u_1^T, \dots, u_{m_t}^T$ of the matrix $\tilde{V}_t^{-1/2} \tilde{\Phi}_t^T$, where $\tilde{\Phi}_t = [\tilde{\varphi}_t(x_1), \dots, \tilde{\varphi}_t(x_t)]^T$ and $\tilde{V}_t = \tilde{\Phi}_t^T \tilde{\Phi}_t + \lambda I_{m_t}$, and use those as the weight vectors for truncating the rewards in each of $m_t$ directions by setting $\hat{r}_i = \sum_{\tau=1}^t u_{i,\tau} y_\tau \mathbb{1}_{|u_{i,\tau} y_\tau| \le b_t}$ for all $i \in [m_t]$, where $b_t$ specifies the truncation level. Then, we find our estimate of $f$ as $\tilde{\theta}_t = \tilde{V}_t^{-1/2} [\hat{r}_1, \dots, \hat{r}_{m_t}]^T$. Finally, we approximate the posterior mean as $\tilde{\mu}_t(x) = \tilde{\varphi}_t(x)^T \tilde{\theta}_t$ and the posterior variance as $(i)$ $\tilde{\sigma}_t^2(x) = \lambda \tilde{\varphi}_t(x)^T \tilde{V}_t^{-1} \tilde{\varphi}_t(x)$ for the Fourier features approximation, or as $(ii)$ $\tilde{\sigma}_t^2(x) = k(x, x) - \tilde{\varphi}_t(x)^T \tilde{\varphi}_t(x) + \lambda \tilde{\varphi}_t(x)^T \tilde{V}_t^{-1} \tilde{\varphi}_t(x)$ for the Nyström approximation. Now it only remains to describe how to find the feature embeddings $\tilde{\varphi}_t$.

**(a) Quadrature Fourier features (QFF) approximation:** If $k$ is a bounded, continuous, positive definite, stationary kernel satisfying $k(x, x) = 1$, then by Bochner's theorem [4], $k$ is the Fourier transform of a probability measure $p$, i.e., $k(x, y) = \int_{\mathbb{R}^d} p(\omega) \cos(\omega^T(x - y)) d\omega$. For the SE kernel, this measure has density $p(\omega) = \left(\frac{l}{\sqrt{2\pi}}\right)^d e^{-\frac{l^2 \|\omega\|_2^2}{2}}$ (abusing notation for measure and density). Mutny and Krause [26] show that for any stationary kernel $k$ on $\mathbb{R}^d$ whose inverse Fourier transform decomposes product wise, i.e., $p(\omega) = \prod_{j=1}^d p_j(\omega_j)$, we can use Gauss-Hermite quadrature [18] to approximate it. If $\mathcal{X} = [0, 1]^d$, the SE kernel is approximated as follows. Choose $\bar{m} \in \mathbb{N}$ and $m = \bar{m}^d$, and construct the $2m$-dimensional feature map

$$\tilde{\varphi}(x)_i = \begin{cases} \sqrt{\nu(\omega_i)} \cos\left(\frac{\sqrt{2}}{l} \omega_i^T x\right) & \text{if } 1 \le i \le m, \\ \sqrt{\nu(\omega_{i-m})} \sin\left(\frac{\sqrt{2}}{l} \omega_{i-m}^T x\right) & \text{if } m + 1 \le i \le 2m. \end{cases} \tag{1}$$

Here the set $\{\omega_1, \dots, \omega_m\} = \overbrace{A_{\bar{m}} \times \cdots \times A_{\bar{m}}}^{d \text{ times}}$, where $A_{\bar{m}}$ is the set of $\bar{m}$ (real) roots of the $\bar{m}$-th Hermite polynomial $H_{\bar{m}}$, and $\nu(z) = \prod_{j=1}^d \frac{2^{\bar{m}-1} \bar{m}!}{\bar{m}^2 H_{\bar{m}-1}(z_j)^2}$ for all $z \in \mathbb{R}^d$. For our purposes, we will have ATA-GP-UCB work with the embedding $\tilde{\varphi}_t(x) = \tilde{\varphi}(x)$ of dimension $m_t = 2m$ for all $t \ge 1$.

***Remark 3.*** The seminal work of Rahimi and Recht [28] that develops random Fourier feature (RFF) approximation of any stationary kernel is based on the feature map $\tilde{\varphi}(x) = \frac{1}{\sqrt{m}} [\cos(\omega_1^T x), \dots, \cos(\omega_m^T x), \sin(\omega_1^T x), \dots, \sin(\omega_m^T x)]^T$, where each $\omega_i$ is sampled independently from $p(\omega)$. However, RFF embeddings do not appear to be useful for our purpose of achieving sublinear regret (see discussion after Lemma 1), so we work with the QFF embedding.

**(b) Nyström approximation:** Unlike the QFF approximation where the basis functions (cosine and sine) do not depend on the data, the basis functions used by the Nyström method are data dependent. For a set of points $\mathcal{X}_t = \{x_1, \dots, x_t\}$, the Nyström method [42] approximates the kernel matrix $K_t$ as follows: First sample a random number $m_t$ of points from $\mathcal{X}_t$ to construct a dictionary $\mathcal{D}_t = \{x_{i_1}, \dots, x_{i_{m_t}}\}; i_j \in [t]$, according to the following distribution. For each $i \in [t]$, include $x_i$ in $\mathcal{D}_t$ independently with probability $p_{t,i} = \min\{q \tilde{\sigma}_{t-1}^2(x_i), 1\}$ for a suitably chosen parameter $q$ (which trades off between the quality and the size of the embedding). Then, compute the (approximate) finite-dimensional feature embedding $\tilde{\varphi}_t(x) = \left(K_{\mathcal{D}_t}^{1/2}\right)^\dagger k_{\mathcal{D}_t}(x)$, where

$K_{\mathcal{D}_t} = [k(u,v)]_{u,v \in \mathcal{D}_t}$, $k_{\mathcal{D}_t}(x) = [k(x_{i_1}, x), \ldots, k(x_{i_{m_t}}, x)]^T$ and $A^\dagger$ denotes the pseudo inverse of any matrix $A$. We call the entire procedure NyströmEmbedding (pseudocode in appendix).

---

**Algorithm 2** Adaptively Truncated Approximate GP-UCB (ATA-GP-UCB)

---

**Input:** Parameters $\lambda > 0$, $\{b_t\}_{t \geq 1}$, $\{\beta_t\}_{t \geq 1}$, $q$, a kernel approximation (QFF or Nyström)
**Set:** $\tilde{\mu}_0(x) = 0$ and $\tilde{\sigma}_0^2(x) = k(x,x)$ for all $x \in \mathcal{X}$
**for** $t = 1, 2, 3 \ldots$ **do**
    Play $x_t = \mathrm{argmax}_{x \in \mathcal{X}} \tilde{\mu}_{t-1}(x) + \beta_t \tilde{\sigma}_{t-1}(x)$ and observe payoff $y_t$
    Set $\tilde{\varphi}_t(x) = \begin{cases} \tilde{\varphi}(x) & \text{if QFF approximation} \\ \text{NyströmEmbedding}\left(\{(x_i, \tilde{\sigma}_{t-1}(x_i))\}_{i=1}^t, q\right) & \text{if Nyström approximation} \end{cases}$
    Set $\tilde{\Phi}_t^T = [\tilde{\varphi}_t(x_1), \ldots, \tilde{\varphi}_t(x_t)]$ and $\tilde{V}_t = \tilde{\Phi}_t^T \tilde{\Phi}_t + \lambda I_{m_t}$, where $m_t$ is the dimension of $\tilde{\varphi}_t$
    Find the rows $u_1^T, \ldots, u_{m_t}^T$ of $\tilde{V}_t^{-1/2} \tilde{\Phi}_t^T$ and set $\hat{r}_i = \sum_{\tau=1}^t u_{i,\tau} y_\tau \mathbb{1}_{|u_{i,\tau} y_\tau| \leq b_t}$ for all $i \in [m_t]$
    Set $\tilde{\theta}_t = \tilde{V}_t^{-1/2} [\hat{r}_1, \ldots, \hat{r}_{m_t}]^T$ and compute $\tilde{\mu}_t(x) = \tilde{\varphi}_t(x)^T \tilde{\theta}_t$
    Set $\tilde{\sigma}_t^2(x) = \begin{cases} (i) \ \lambda \tilde{\varphi}_t(x)^T \tilde{V}_t^{-1} \tilde{\varphi}_t(x) & \text{if QFF approximation} \\ (ii) \ k(x,x) - \tilde{\varphi}_t(x)^T \tilde{\varphi}_t(x) + \lambda \tilde{\varphi}_t(x)^T \tilde{V}_t^{-1} \tilde{\varphi}_t(x) & \text{if Nyström approximation} \end{cases}$
**end for**

---

**Remark 4.** It is well known ($\lambda$-ridge leverage score sampling [1]) that, by sampling points proportional to their posterior variances $\sigma_t^2(x)$, one can obtain an accurate embedding $\tilde{\varphi}_t(x)$, which in turn gives an accurate approximation $\tilde{\sigma}_t^2(x)$. But, computation of $\sigma_t^2(x)$ in turn requires inverting $K_t$, which takes at most $O(t^3)$ time. So, we make use of the already computed approximations $\tilde{\sigma}_{t-1}^2(x)$ to sample points at round $t$, without significantly compromising on the accuracy of the embeddings [9].

**Remark 5.** The choice $(i)$ of $\tilde{\sigma}_t^2(x)$ in Algorithm 2 ensures accurate estimation of the variance of $x$ under the QFF approximation [26]. But, the same choice leads to severe underestimation of the variance under the Nyström approximation, specially when $x$ is far away from $\mathcal{D}_t$. The choice $(ii)$ of $\tilde{\sigma}_t^2(x)$ in Algorithm 2 is known as *deterministic training conditional* in the GP literature [27] and provably prevents the phenomenon of variance starvation under Nyström approximation [9].

### 5.2 Cumulative regret of ATA-GP-UCB with QFF embeddings

The following lemma shows that the data adaptive truncation of all the historical rewards and a good approximation of the kernel help us obtain a tighter confidence interval than TGP-UCB.

**Lemma 1 (Tighter confidence sets with QFF truncation)** *For any $\delta \in (0,1]$, ATA-GP-UCB with QFF approximation and parameters $b_t = (v/\ln(2mT/\delta))^{\frac{1}{1+\alpha}} t^{\frac{1-\alpha}{2(1+\alpha)}}$ and $\beta_{t+1} = B + 4\sqrt{m/\lambda}\ v^{\frac{1}{1+\alpha}} (\ln(2mT/\delta))^{\frac{\alpha}{1+\alpha}} t^{\frac{1-\alpha}{2(1+\alpha)}}$, ensures that with probability at least $1 - \delta$, uniformly over all $t \in [T]$ and $x \in \mathcal{X}$,*

$$|f(x) - \tilde{\mu}_{t-1}(x)| \leq \beta_t \tilde{\sigma}_{t-1}(x) + O(B\varepsilon_m^{1/2} t^2), \tag{2}$$

*where the QFF dimension $m$ is such that $\sup_{x,y \in \mathcal{X}} \left|k(x,y) - \tilde{\varphi}(x)^T \tilde{\varphi}(y)\right| =: \varepsilon_m < 1$.*

Here, the scaling $t^{\frac{1-\alpha}{2(1+\alpha)}}$ of the confidence width $\beta_t$ is much less than the scaling $t^{\frac{1}{2(1+\alpha)}}$ of TGP-UCB, which eventually leads to a tighter confidence interval. However, in order to achive sublinear cumulative regret, we need to ensure that the approximation error $\varepsilon_m$ decays at least as fast as $O(1/T^6)$ and feature dimension $m$ grows no faster than polylog$(T)$. This will ensure that the regret accumulated due to the second term in the RHS of 2 is $O(1)$, as well as the contribution from the first term is $\tilde{O}(T^{\frac{1}{1+\alpha}})$, since sum of the approximate posterior standard deviations grows only as $\tilde{O}(\sqrt{mT})$. Now, the QFF embedding (1) of $k_{\mathrm{SE}}$ can be shown to achieve $\varepsilon_m \leq d2^{d-1} \frac{1}{\sqrt{2\bar{m}\bar{m}}} \left(\frac{e}{4l^2}\right)^{\bar{m}} = O\left(\frac{d2^{d-1}}{(\bar{m}l^2)^{\bar{m}}}\right)$ [26]. The decay is exponential when $\bar{m} > 1/l^2$ and $d = O(1)^2$. Now, setting $\bar{m} = \Theta\left(\log_{4/e}\left(T^6\right)\right)$, we can ensure that $\varepsilon_m^{1/2} T^3 = O(1)$ and $m = O\left((\ln T)^d\right)$, and thus, in turn, a sublinear regret bound [3]. The following theorem states this formally, with a full proof deferred to Appendix D.2.

**Theorem 3 (Regret bound for ATA-GP-UCB with QFF embedding)** *Fix any $\delta \in (0,1]$. Then, under the same hypothesis of Theorem 1, for $\mathcal{X} = [0,1]^d$ and $k = k_{SE}$, ATA-GP-UCB under QFF approximation, with parameters $b_t$ and $\beta_t$ set as in Lemma 1, and with the embedding $\tilde{\varphi}$ from 1 such that $\bar{m} > 1/l^2$ and $\bar{m} = \Theta\left(\log_{4/e}\left(T^6\right)\right)$, enjoys, with probability at least $1 - \delta$, the regret bound*

$$R_T = O\left(B\sqrt{T(\ln T)^{d+1}} + v^{\frac{1}{1+\alpha}}\left(\ln\left(\frac{T(\ln T)^d}{\delta}\right)\right)^{\frac{\alpha}{1+\alpha}}\sqrt{\ln T}\,(\ln T)^d\,T^{\frac{1}{1+\alpha}}\right).$$

***Remark 6.*** When the variance of the rewards is finite (i.e., $\alpha = 1$), the cumulative regret for ATA-GP-UCB under QFF approximation of the SE kernel is $O((\ln T)^{d+1}\sqrt{T})$, which now recovers the state-of-the-art regret bound of GP-UCB under sub-Gaussian rewards [26, Corollary 2] unlike the earlier TGP-UCB. It is worth pointing out that the bound in Theorem 3 is only for the SE kernel defined on $\mathcal{X} = [0,1]^d$, and designing a no-regret BO strategy under the QFF approximation of any other stationary kernel still remains a open question even when the rewards are sub-Gaussian [26].

### 5.3 Cumulative regret of ATA-GP-UCB with Nyström embeddings

Now, we will show that ATA-GP-UCB under Nyström approximation achives optimal regret for any stationary kernel defined on $\mathcal{X} \subset \mathbb{R}^d$ without any restriction on $d$. Similar to Lemma 1, ATA-GP-UCB under Nyström approximation also maintains tighter confidence sets than TGP-UCB. As before, the confidence sets are useful only if the dimension of the embeddings $m_t$ grows no faster than polylog$(t)$. Not only that, we also need to ensure that the approximate posterior variances are only a constant factor away from the exact ones. Then, since sum of the posterior standard deviations grows only as $O(\sqrt{T\gamma_T})$, we can achieve the optimal $\tilde{O}(T^{\frac{1}{1+\alpha}})$ regret scaling. Now for any $\varepsilon \in (0,1)$, setting $q = 6\frac{1+\varepsilon}{1-\varepsilon}\ln(2T/\delta)/\varepsilon^2$, the Nyström embeddings $\tilde{\varphi}_t$ can be shown to achieve $m_t \leq 6\frac{1+\varepsilon}{1-\varepsilon}\left(1+\frac{1}{\lambda}\right)q\gamma_t$ and $\frac{1-\varepsilon}{1+\varepsilon}\sigma_t^2(x) \leq \tilde{\sigma}_t^2(x) \leq \frac{1+\varepsilon}{1-\varepsilon}\sigma_t^2(x)$ with probability at least $1 - \delta$ [9], which helps us to achieve an optimal regret bound. The following theorem states this formally, with a full proof deferred to Appendix D.3.

**Theorem 4 (Regret bound for ATA-GP-UCB with Nyström embedding)** *Fix any $\delta \in (0,1]$, $\varepsilon \in (0,1)$ and set $\rho = \frac{1+\varepsilon}{1-\varepsilon}$. Then, under the same hypothesis of Theorem 1, ATA-GP-UCB under Nyström approximation, and with parameters $q = 6\rho\ln(4T/\delta)/\varepsilon^2$, $b_t = (v/\ln(4m_tT/\delta))^{\frac{1}{1+\alpha}}t^{\frac{1-\alpha}{2(1+\alpha)}}$ and $\beta_{t+1} = B(1 + \frac{1}{\sqrt{1-\varepsilon}}) + 4\sqrt{m_t/\lambda}\,v^{\frac{1}{1+\alpha}}(\ln(4m_tT/\delta))^{\frac{\alpha}{1+\alpha}}t^{\frac{1-\alpha}{2(1+\alpha)}}$, enjoys, with probability at least $1 - \delta$, the regret bound*

$$R_T = O\left(\rho B\left(1 + \frac{1}{\sqrt{1-\varepsilon}}\right)\sqrt{T\gamma_T} + \frac{\rho^2}{\varepsilon}v^{\frac{1}{1+\alpha}}\left(\ln\left(\frac{\gamma_T\ln(T/\delta)T}{\delta}\right)\right)^{\frac{\alpha}{1+\alpha}}\sqrt{\ln(T/\delta)}\gamma_T T^{\frac{1}{1+\alpha}}\right).$$

***Remark 7.*** Theorems 3 and 4 imply that ATA-GP-UCB achieves $\tilde{O}\left(v^{\frac{1}{1+\alpha}}(\ln T)^d T^{\frac{1}{1+\alpha}}\right)$ regret bound for $k_{SE}$, which matches the lower bound (Theorem 2) upto a factor of $\frac{\alpha}{1+\alpha}$ in the exponent of $\ln T$, as well as a few extra $\ln T$ factors hidden in the notation $\tilde{O}$. For the Matérn kernel, the bound is $\tilde{O}\left(T^{\frac{1}{1+\alpha}\frac{2\nu+(2+\alpha)d(d+1)}{2\nu+d(d+1)}}\right)$, which is sublinear only when $\frac{d(d+1)}{2\nu} < \alpha^4$, and the gap from the lower bound is more significant in this case. It is worth mentioning that a similar gap is present even for the (easier) setting of sub-Gaussian rewards [31] and there might exist better algorithms which can bridge this gap. When the variance of the rewards is finite (i.e., $\alpha = 1$), the cumulative regret for ATA-GP-UCB under Nyström approximation is $\tilde{O}(\gamma_T\sqrt{T})$, which recovers the state-of-the-art regret bound under sub-Gaussian rewards [9, Thm. 2]. For the linear bandit setting, i.e. when the feature map $\tilde{\varphi}_t(x) = x$ itself, substituting $\gamma_T = O(d\ln T)$, we find that the regret upper bound in Theorem 4 recovers the (optimal) regret bound of [34, Thm. 3] up to a logarithmic factor.

### 5.4 Computational complexity of ATA-GP-UCB

**(a) Time complexity:** Under the (data-dependent) Nyström approximation, constructing the dictionary $D_t$ takes $O(t)$ time at each step $t$. Then, we compute the embeddings $\tilde{\varphi}_t(x)$ for all arms in

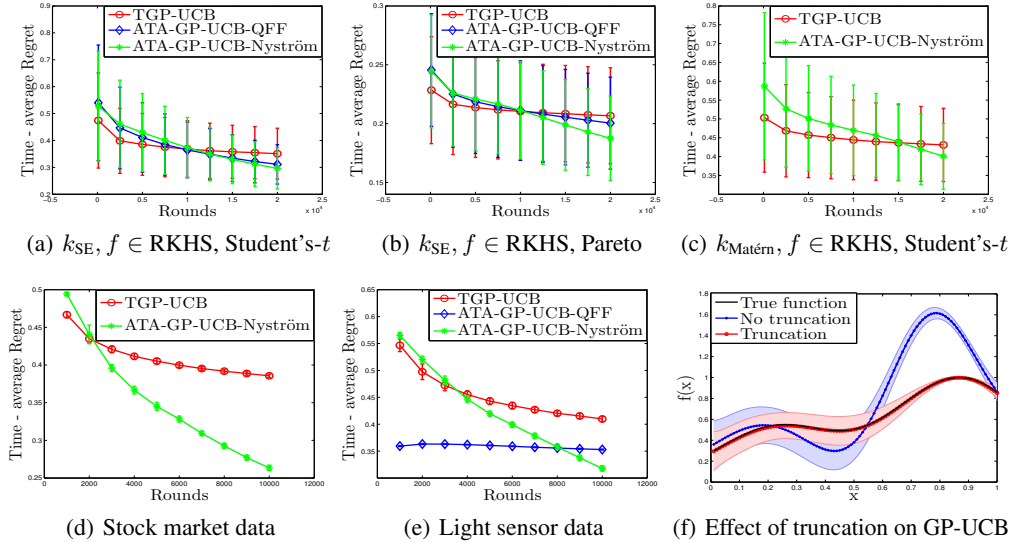

Figure 1: (a)-(e) Time-average regret ($R_T/T$) for TGP-UCB, ATA-GP-UCB with QFF approximation (ATA-GP-UCB-QFF) and Nyström approximation (ATA-GP-UCB-Nyström) on heavy-tailed data. (f) Confidence sets ($\mu_t \pm \sigma_t$) formed by GP-UCB with and without truncation under heavy fluctuations.

$O(m_t^3 + m_t^2 |\mathcal{X}|)$ time, where $|\mathcal{X}|$ is the cardinality of $\mathcal{X}$. Now, construction of $\tilde{V}_t$ takes $O(m_t^2 t)$ time, since we need to rebuild it from the scratch. Then, $\tilde{V}_t^{-1/2}$ is computed in $O(m_t^3)$ time. We can now compute $\tilde{\mu}_t(x)$ and $\tilde{\sigma}_t^2(x)$ for all arms in $O(m_t^2 t + m_t |\mathcal{X}|)$ and $O(m_t^2 |\mathcal{X}|)$ time, respectively, using already computed $\tilde{\varphi}_t(x)$ and $\tilde{V}_t^{-1/2}$. Thus per-step time complexity is $O\left(m_t^2(t + |\mathcal{X}|)\right)$, since $m_t \leq t$. For continuous $\mathcal{X}$, one can approximately maximize the GP-UCB score by grid search / Branch and Bound methods such as DIRECT [6]. In fact it can be maximized within $O(\varepsilon)$ accuracy by making $O(\varepsilon^{-d})$ calls to it, yielding a per-step time complexity of $O(m_t^2(t + \varepsilon^{-d}))$. Since $m_t = \tilde{O}(\gamma_t)$ and $\gamma_t$ is poly-logarithmic in $t$ for SE kernel, per step time complexity is $\tilde{O}(t + \varepsilon^{-d})$. For Matérn kernel, the complexity is $\tilde{O}(t^p(t + \varepsilon^{-d})), 1 < p < 2$. Similarly, under (data-independent) QFF approximation, the per-step time complexity is $O(m^3 + m^2(t + \varepsilon^{-d})) = \tilde{O}(t + \varepsilon^{-d})$ since $m = O((\ln T)^d)$ for the SE kernel.

**(b) Space complexity:** Note that under Nyström approximation, at each round $t$ we need to store all previously chosen arms, the matrix $\tilde{V}_t^{-1/2}$ and the vectors $\tilde{\varphi}_t(x)$. Hence, the per-step space complexity of ATA-GP-UCB is $O(t + m_t(m_t + \varepsilon^{-d})) = O(m_t(m_t + \varepsilon^{-d}))$ for small enough $\varepsilon$. Under QFF approximation, the complexity is $O(m(m + \varepsilon^{-d}))$.

## 6 Experiments

We numerically compare the performance of TGP-UCB (Algorithm 1), ATA-GP-UCB with QFF (ATA-GP-UCB-QFF) and Nyström (ATA-GP-UCB-Nyström) approximations (Algorithm 2) on both synthetic and real-world heavy-tailed environments. (Our codes are available here.) The confidence width $\beta_t$ and truncation level $b_t$ of our algorithms, and the trade-off parameter $q$ used in Nyström approximation are set order-wise similar to those recommended by theory (Theorems 1, 3 and 4). We use $\lambda = 1$ in all algorithms and $\varepsilon = 0.1$ in ATA-GP-UCB-Nyström. We plot the mean and standard deviation (under independent trials) of the time-average regret $R_T/T$ in Figure 1. We use the following datasets.

**1. Synthetic data:** We generate the objective function $f \in \mathcal{H}_k(\mathcal{X})$ with $\mathcal{X}$ set to be a discretization of $[0, 1]$ into 100 evenly spaced points. Each $f = \sum_{i=1}^{p} a_i k(\cdot, x_i)$ was generated using an SE kernel with $l = 0.2$ and by uniformly sampling $a_i \in [-1, 1]$ and support points $x_i \in \mathcal{X}$ with $p = 100$. We set $B = \max_{x \in \mathcal{X}} |f(x)|$. To generate the rewards, first we consider $y(x) = f(x) + \eta$, where the noise $\eta$ are samples from the Student's $t$-distribution with 3 degrees of freedom (Figure 1 a). Here, the

variance is bounded ($\alpha = 1$) and hence $v = B^2 + 3$. Next, we generate the rewards as samples from the Pareto distribution with shape parameter 2 and scale parameter $f(x)/2$. $f$ is generated similarly, except that here we sample $a_i$'s uniformly from $[0, 1]$. Then, we set $B$ as before leading to the bound of $(1 + \alpha)$-th raw moments $v = \frac{B^{1+\alpha}}{2^\alpha(1-\alpha)}$. We plot the results for $\alpha = 0.9$ (Figure 1 b). We use $m = 32$ features (in consistence with Theorem 3) for ATA-GP-UCB-QFF in these experiments. Next, we generate $f$ using the Matérn kernel with $l = 0.2$ and $\nu = 2.5$, and consider the same Student's-$t$ distribution as earlier to generate rewards. As we do not have the theory of ATA-GP-UCB-QFF for the Matérn kernel yet, we exclude evaluating it here (Figure 1 c). We perform 20 trials for $2 \times 10^4$ rounds and for each trial we evaluate on a different $f$ (which explains the high error bars).

**2. Stock market data:** We consider a representative application of identifying the most profitable stock in a given pool of stocks. This is motivated by the practical scenario that an investor would like to invest a fixed budget of money in a stock and get as much return as possible. We took the adjusted closing price of 29 stocks from January 4th, 2016 to April 10th, 2019. We conduct Kolmogrov-Smirnov (KS) test to find out that the null hypothesis of stock prices following a Gaussian distribution is rejected against the favor of a heavy-tailed distribution. We take the empirical mean of stock prices as our objective function $f$ and empirical covariance of the normalized stock prices as our kernel function $k$ (since stock behaviors are mostly correlated with one another). We consider $\alpha = 1$ and set $v$ as the empirical average of the squared prices. Since the kernel is data dependent, we cannot run ATA-GP-UCB-QFF here. We average over 10 independent trials of the algorithms (Figure 1 d).

**3. Light sensor data:** We take light sensor data collected in the CMU Intelligent Workplace in Nov 2005 containing locations of 41 sensors, 601 train samples and 192 test samples in the context of learning the maximum average reading of the sensors. For each sensor, we find that the KS test on its readings rejects the Gaussian against the favor of a heavy-tailed distribution. We take the empirical average of the test samples as our objective $f$ and empirical covariance of the normalized train samples as our kernel $k$. We consider $\alpha = 1$, set $v$ as the empirical mean of the squared readings and $B$ as the maximum of the average readings. For ATA-GP-UCB-QFF, we fit a SE kernel with $l^2 = 0.1$ on the given sensor locations and approximate it with $m = 16^2 = 256$ features (Figure 1 e).

**Observations:** We find that ATA-GP-UCB outperforms TGP-UCB uniformly over all experiments, which is consistent with our theoretical results. We also see that the performance of ATA-GP-UCB under the Nyström approximation is no worse than that under the QFF approximation. Not only that, the scope of the latter is limited due to its dependence on the analytical form of the kernel, whereas the former is data-adaptive and hence, well suited for practical purposes.

**Effect of truncation:** For heavy-tailed rewards, the sub-Gaussian constant $R = \infty$. Hence, we exclude evaluating GP-UCB in the above experiments. Now, we demonstrate the effect of truncation on GP-UCB in the following experiment. First, we generate a function $f \in \mathcal{H}_k(\mathcal{X})$ and normalize it between $[0, 1]$. Then, we simulate rewards as $y(x) = f(x) + \eta$, where $\eta$ takes values in $\{-10, 10\}$, uniformly, for any single random point in $\mathcal{X}$, and is zero everywhere else. We run GP-UCB with $\beta_t = \ln t$ and see that the posterior mean after $T = 10^4$ rounds is not a good estimate of $f$. However, by truncating reward samples which exceeds $t^{1/4}$ (truncation threshold in TGP-UCB when $\alpha = 1$) at round $t$, we get an (almost) accurate estimator of $f$. Not only that, the confidence interval around this estimator contains $f$ at every point in $\mathcal{X}$, which in turn ensures good performance. We plot the respective confidence sets averaged over 50 such randomizations of noise (Figure 1 f).

## 7 Conclusion

To the best of our knowledge, this is the first work to formulate and solve BO under heavy-tailed observations. We have demonstrated the failure of existing BO methods and developed (almost) optimal algorithms using kernel approximation techniques, which are easy to implement and perform well in practice, with rigorous theoretical guarantees. It is worth noting that using a Bernstein type concentration bound in each direction of the approximate feature space, we are able to obtain the near optimal regret scaling for ATA-GP-UCB (Algorithm 2). Instead, if one can derive a Bernstein type bound for self-normalized processes which depends on the $(1 + \alpha)$-th moments of rewards, then one may not have to resort to feature approximation to get optimal regret. Further, instead of truncating the payoffs, one can also consider building and studying a median of means-style estimator [8] in the (approximate) feature space and hope to develop an optimal algorithm.

**Acknowledgments**

The authors are grateful to the anonymous reviewers for their valuable comments. S. R. Chowdhury is supported by the Google India PhD fellowship grant and the Tata Trusts travel grant. A. Gopalan is grateful for support from the DST INSPIRE faculty grant IFA13- ENG-69.

## Footnotes

[1]If instead the moment bound holds for each $\eta_t$ then this can be translated to a moment bound for each $y_t$ using, say, a bound on $f(x)$.

[2] For most BO applications, the effective dimensionality of the problem is low, e.g., additive models [21, 30].

[3] Under RFF approximation $\varepsilon_m = \tilde{O}(\sqrt{1/m})$ [36]. Hence, ATA-GP-UCB does not achieve sublinear regret.

[4]This holds, for example, Matérn kernel on $\mathbb{R}^2$ with $\nu = 3.5$ when variance of the rewards is finite ($\alpha = 1$).

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
