[Supplementary Material · Supplementary.pdf]

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

[5]In our setting, $B$ and $v$ are constants that do not scale with $T$ and the condition is trivially satisfied.

[6]For the RFF approximation, we have $\varepsilon_m = O_p(1/\sqrt{m})$ if $d = O(1)$. Now in order to make the last term $\varepsilon_m^{1/2} T^3$ behave as $O(1)$, we have to take $m = O(T^{12})$ features which will eventually blow up the first two terms by the same order. Hence, we will never achieve sub-linear regret bound using RFF approximation.

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

# Appendix

## A  Preliminaries

First, we review some useful matrix identities.

**Lemma 2** *[16, Lemma 12] Let $A \succeq B \succ 0$ be positive definite matrices. Then $A^{-1} \bullet (A - B) = \ln \frac{|A|}{|B|}$, where $X \bullet Y := \sum_{i=1}^{n} \sum_{j=1}^{n} X_{i,j} Y_{i,j}$ for any two matrices $X, Y \in \mathbb{R}^{n \times n}$.*

**Lemma 3** *For any linear operator $A : \mathcal{H}_k(\mathcal{X}) \to \mathbb{R}^t$ and its adjoint $A^T : \mathbb{R}^t \to \mathcal{H}_k(\mathcal{X})$, and for any $\lambda > 0$,*

$$(A^T A + \lambda I_{\mathcal{H}})^{-1} A^T = A^T (AA^T + \lambda I_t)^{-1}, \tag{3}$$

*and*

$$I_{\mathcal{H}} - A^T (AA^T + \lambda I_t) A = \lambda (A^T A + \lambda I_{\mathcal{H}})^{-1}. \tag{4}$$

**Proof**  The proofs follow from the fact that $(A^T A + \lambda I_{\mathcal{H}}) A^T = A^T (AA^T + \lambda I_t)$ for any $\lambda > 0$. ∎

Next, we review some relevant definitions and results, which will be useful in the analysis of our algorithms. We first begin with the definition of *Maximum Information Gain*, first appeared in [35], which basically measures the reduction in uncertainty about the unknown function after some noisy observations (rewards).

For a function $f : \mathcal{X} \to \mathbb{R}$ and any subset $A \subset \mathcal{X}$ of its domain, we use $f_A := [f(x)]_{x \in A}$ to denote its restriction to $A$, i.e., a vector containing $f$'s evaluations at each point in $A$ (under an implicitly understood bijection from coordinates of the vector to points in $A$). In case $f$ is a random function, $f_A$ will be understood to be a random vector. For jointly distributed random variables $X$ and $Y$, $I(X; Y)$ denotes the Shannon mutual information between them.

**Definition 1 (Maximum Information Gain (MIG))** *Let $f : \mathcal{X} \to \mathbb{R}$ be a (possibly random) real-valued function defined on a domain $\mathcal{X}$, and $t$ a positive integer. For each subset $\mathcal{X} \subset D$, let $Y_A$ denote a noisy version of $f_A$ obtained by passing $f_A$ through a channel $\mathbb{P}[Y_A | f_A]$. The Maximum Information Gain (MIG) about $f$ after $t$ noisy observations is defined as*

$$\gamma_t := \max_{A \subset \mathcal{X} : |A| = t} I(f_A; Y_A).$$

*(We omit mentioning explicitly the dependence on the channels for ease of notation.)*

Let $k : \mathcal{X} \times \mathcal{X} \to \mathbb{R}$ be a symmetric positive semi-definite kernel and for any $A \subset \mathcal{X}$, let $K_A$ denotes the induced kernel matrix.

**Lemma 4 (MIG under GP prior and additive Gaussian noise [35])** *Let $f \sim GP_{\mathcal{X}}(0, k)$ be a sample from a Gaussian process over $\mathcal{X}$ and $Y_A$ denote a noisy version of $f_A$ obtained by passing $f_A$ through a channel that adds iid $\mathcal{N}(0, \lambda)$ noise to each element of $f_A$. Then,*

$$\gamma_t \equiv \gamma_t(k, \mathcal{X}) = \max_{A \subset \mathcal{X} : |A| = t} \frac{1}{2} \ln \left| I + \lambda^{-1} K_A \right|.$$

Srinivas et al. [35] proved upper bounds over $\gamma_t$ for commonly used kernels. The bounds are given in Lemma 5.

**Lemma 5 (MIG for common kernels [35])** *Let $\mathcal{X}$ be a compact and convex subset of $\mathbb{R}^d$ and the kernel $k$ satisfies $k(x, x') \le 1$ for all $x, x' \in \mathcal{X}$. Then for*

- *Linear kernel: $\gamma_t = O(d \ln t)$.*

- *Squared Exponential kernel: $\gamma_t = O\left((\ln t)^{d+1}\right)$.*

- *Matérn kernel: $\gamma_t = O\left(t^{\frac{d(d+1)}{2\nu + d(d+1)}} \ln t\right)$.*

Note that, MIG depends only *sublinearly* on the number of observations $t$ for all these kernels and it will serve as a key instrument to obtain our regret bounds by virtue of Lemma 4 and 6.

Now, observe that any kernel function $k : \mathcal{X} \times \mathcal{X} \to \mathbb{R}, \mathcal{X} \subset \mathbb{R}^d$ is associated with a non-linear feature map $\varphi : \mathcal{X} \to \mathcal{H}_k(\mathcal{X})$ such that $k(x, y) = \langle \varphi(x), \varphi(y) \rangle_{\mathcal{H}}$, where $\langle \cdot, \cdot \rangle_{\mathcal{H}}$ denotes the inner product in the RKHS $\mathcal{H}_k(\mathcal{X})$ and $\|\cdot\|_{\mathcal{H}}$ denotes the corresponding norm. Observe that for any $h \in \mathcal{H}_k(\mathcal{X}), h(x) = \langle h, \varphi(x) \rangle_{\mathcal{H}}$ by the reproducing property. For a set $\{x_1, \ldots, x_t\} \subset \mathcal{X}$ define the operator $\Phi_t : \mathcal{H}_k(\mathcal{X}) \to \mathbb{R}^t$ such that for any $h \in \mathcal{H}_k(\mathcal{X}), \Phi_t h = [\langle \varphi(x_1), h \rangle_{\mathcal{H}}, \ldots, \langle \varphi(x_t), h \rangle_{\mathcal{H}}]^T$, and denote its adjoint by $\Phi_t^T : \mathbb{R}^t \to \mathcal{H}_k(\mathcal{X})$. By reproducing property $\varphi_t h = [h(x_1), \ldots, h(x_t)]^T$. For any $\lambda > 0$, define $V_t = \Phi_t^T \Phi_t + \lambda I_{\mathcal{H}}$, where $I_{\mathcal{H}} : \mathcal{H}_k(\mathcal{X}) \to \mathcal{H}_k(\mathcal{X})$ denotes the identity operator. For a positive definite operator $V : \mathcal{H}_k(\mathcal{X}) \to \mathcal{H}_k(\mathcal{X})$, define the inner product $\langle \cdot, \cdot \rangle_V := \langle \cdot, V \cdot \rangle_{\mathcal{H}}$ with corresponding norm $\|\cdot\|_V$. Observe that, under this definition, the posterior variance $\sigma_t^2(x) = \lambda \|\varphi(x)\|_{V_t^{-1}}^2$.

**Lemma 6 (Sum of predictive variances and MIG)** *If $k(x, x) \leq 1$ for all $x \in \mathcal{X}$, then*

$$\sum_{s=1}^{t} \sigma_{s-1}^2(x_s) \leq 2(1 + \lambda)\gamma_t.$$

**Proof** Observe that $V_t = V_{t-1} + \varphi(x_t)\varphi(x_t)^T$. Therefore, by Sherman–Morrison-Woodbury matrix identity, we have $V_t^{-1} = V_{t-1}^{-1} - \frac{V_{t-1}^{-1}\varphi(x_t)\varphi(x_t)^T V_{t-1}^{-1}}{1 + \varphi(x_t)^T V_{t-1}^{-1}\varphi(x_t)}$. This, in turn, implies that

$$\|\varphi(x)\|_{V_t^{-1}}^2 = \|\varphi(x)\|_{V_{t-1}^{-1}}^2 - \frac{\langle \varphi(x), \varphi(x_t) \rangle_{V_{t-1}^{-1}}^2}{1 + \|\varphi(x_t)\|_{V_{t-1}^{-1}}^2} \overset{(a)}{\geq} \|\varphi(x)\|_{V_{t-1}^{-1}}^2 \left( 1 - \frac{\|\varphi(x_t)\|_{V_{t-1}^{-1}}^2}{1 + \|\varphi(x_t)\|_{V_{t-1}^{-1}}^2} \right) = \frac{\|\varphi(x)\|_{V_{t-1}^{-1}}^2}{1 + \|\varphi(x_t)\|_{V_{t-1}^{-1}}^2}$$

where $(a)$ follows from Cauchy-Schwartz inequality. Since $V_{t-1} \succeq \lambda I_{\mathcal{H}}$, we have $\|\varphi(x_t)\|_{V_{t-1}^{-1}}^2 \leq \frac{1}{\lambda}\|\varphi(x_t)\|_{\mathcal{H}}^2 = \frac{1}{\lambda}k(x_t, x_t) \leq \frac{1}{\lambda}$. This implies that $\|\varphi(x)\|_{V_{t-1}^{-1}}^2 \leq (1 + \frac{1}{\lambda})\|\varphi(x)\|_{V_t^{-1}}^2$ and therefore

$$\sigma_{t-1}^2(x) \leq \left( 1 + \frac{1}{\lambda} \right) \sigma_t^2(x) \text{ for all } x \in \mathcal{X}. \tag{5}$$

Observe that $\varphi(x_t)^T V_t^{-1}\varphi(x_t) = V_t^{-1} \bullet \varphi(x_t)\varphi(x_t)^T = V_t^{-1} \bullet (V_t - V_{t-1})$ since for any $a \in \mathbb{R}^n$ and $B \in \mathbb{R}^{n \times n}, a^T B a = B \bullet a a^T$. Then from Lemma 2, we have $\frac{1}{\lambda}\sigma_t^2(x_t) = \ln \frac{|V_t|}{|V_{t-1}|}$ and thus, in turn,

$$\frac{1}{\lambda} \sum_{s=1}^{t} \sigma_s^2(x_s) \leq \ln \frac{|V_t|}{|V_0|} = \ln \left| \lambda^{-1}\Phi_t^T \Phi_t + I_{\mathcal{H}} \right| = \ln \left| \lambda^{-1}\Phi_t \Phi_t^T + I_t \right| = \ln \left| \lambda^{-1}K_t + I_t \right|. \tag{6}$$

Combining 5 and 6, we get

$$\sum_{s=1}^{t} \sigma_{s-1}^2(x_s) \leq \left( 1 + \frac{1}{\lambda} \right) \sum_{s=1}^{t} \sigma_s^2(x_s) \leq (1 + \lambda)\ln \left| \lambda^{-1}K_t + I_t \right|.$$

Now the result follows from Lemma 4. ∎

# B   Analysis of TGP-UCB

The following lemma states a self-normalized concentration inequality for RKHS-valued martingales.

**Lemma 7 (RKHS-valued martingale control [13])** *Let $\{z_t\}_{t \geq 1}$ be an $\mathbb{R}^d$-valued discrete time stochastic processes such that $z_t$ is predictable with respect to a filtration $\{\mathcal{G}_t\}_{t \geq 0}$, i.e., $z_t$ is $\mathcal{G}_{t-1}$-measurable for all $t \geq 1$. Let $\{w_t\}_{t \geq 1}$ be a real-valued stochastic process such that for all $t \geq 1, w_t$*

*is (a) $\mathcal{G}_t$-measurable, and (b) R-sub-Gaussian conditionally on $\mathcal{G}_{t-1}$ for some $R > 0$. Then, for any $\delta \in (0,1]$, with probability at least $1 - \delta$, uniformly over all $t \geq 1$,*

$$\left\| \sum_{\tau=1}^{t} w_\tau \varphi(z_\tau) \right\|_{Z_t^{-1}} \leq R \sqrt{2 \left( \frac{1}{2} \ln \frac{|Z_t|}{|Z|} + \ln(1/\delta) \right)}.$$

*where $Z_t = Z + \sum_{\tau=1}^{t} \varphi(z_\tau)\varphi(z_\tau)^T$ and $Z : \mathcal{H}_k(\mathbb{R}^d) \to \mathcal{H}_k(\mathbb{R}^d)$ is a positive definite operator.*

Observe that $\sum_{\tau=1}^{t} w_\tau \varphi(z_\tau)$ is $\mathcal{G}_t$-measurable and $\mathbb{E}\left[ \sum_{\tau=1}^{t} w_\tau \varphi(z_\tau) | \mathcal{G}_{t-1} \right] = \sum_{\tau=1}^{t-1} w_\tau \varphi(z_\tau)$. The process $\left( \sum_{\tau=1}^{t} w_\tau \varphi(z_\tau) \right)_{t \geq 1}$ is thus a martingale with respect to the filtration $(\mathcal{G}_t)_{t \geq 0}$ with values in the RKHS $\mathcal{H}_k(\mathcal{X})$, whose deviation is measured by the norm weighted by $Z_t^{-1}$, which is derived from the process itself. Hence, the name self-normalized concentration inequality. Now, we will show that $f$ lies in the confidence sets constructed by TGP-UCB with high probability.

**Lemma 8 (Confidence sets of TGP-UCB contains $f$)** *Let $f \in \mathcal{H}_k(\mathcal{X})$, $\|f\|_{\mathcal{H}} \leq B$ and $k(x,x) \leq 1$ for all $x \in \mathcal{X}$. Let $\mathbb{E}\left[ |y_t|^{1+\alpha} | \mathcal{F}_{t-1} \right] \leq v < \infty$ for some $\alpha \in (0,1]$ and for all $t \geq 1$. Then, for any $\delta \in (0,1]$, TGP-UCB, with $b_t = v^{\frac{1}{1+\alpha}} t^{\frac{1}{2(1+\alpha)}}$ and $\beta_{t+1} = B + \frac{3}{\sqrt{\lambda}} v^{\frac{1}{1+\alpha}} t^{\frac{1}{2(1+\alpha)}} \sqrt{\ln |I_t + \lambda^{-1} K_t| + 2\ln(1/\delta)}$, ensures, with probability at least $1 - \delta$, uniformly over all $x \in \mathcal{X}$ and $t \geq 1$, that*

$$|f(x) - \widehat{\mu}_{t-1}(x)| \leq \beta_t \sigma_{t-1}(x).$$

**Proof** First, we define $\alpha_t(x) = k_t(x)^T (K_t + \lambda I_t)^{-1} f_t$, where $f_t = [f(x_1), \ldots, f(x_t)]^T$ is a vector containing $f$'s evaluations up to round $t$. By reproducing property, $\alpha_t(x) = \langle \varphi(x), \Phi_t^T (\Phi_t \Phi_t^T + \lambda I_t)^{-1} \Phi_t f \rangle_{\mathcal{H}}$. Then, we have

$$f(x) - \alpha_t(x) = \langle \varphi(x), \left( I_{\mathcal{H}} - \Phi_t^T (\Phi_t \Phi_t^T + \lambda I_t)^{-1} \Phi_t \right) f \rangle_{\mathcal{H}} \stackrel{(a)}{=} \lambda \langle \varphi(x), f \rangle_{V_t^{-1}} = \lambda \langle V_t^{-1/2} \varphi(x), V_t^{-1/2} f \rangle_{\mathcal{H}},$$

where $(a)$ follows from 4. By Cauchy-Schwartz inequality, we have for any $x \in \mathcal{X}$

$$\begin{aligned} |f(x) - \alpha_t(x)| &\leq \lambda \left\| V_t^{-1/2} \varphi(x) \right\|_{\mathcal{H}} \left\| V_t^{-1/2} f \right\|_{\mathcal{H}} \\ &\stackrel{(a)}{\leq} \lambda^{1/2} \|\varphi(x)\|_{V_t^{-1}} \|f\|_{\mathcal{H}} \stackrel{(b)}{\leq} B \, \sigma_t(x). \end{aligned} \tag{7}$$

Here in $(a)$ we have used the fact that $V_t^{-1} \preceq \lambda^{-1} I_{\mathcal{H}}$, and hence, $\left\| V_t^{-1/2} f \right\|_{\mathcal{H}} \leq \lambda^{-1/2} \|f\|_{\mathcal{H}}$. $(b)$ follows from $\|f\|_{\mathcal{H}} \leq B$. Now, let $\widehat{\eta}_t = \widehat{y}_t - f(x_t), t = 1, 2, \ldots$ denotes the truncated noise and $\widehat{N}_t = [\widehat{\eta}_1, \ldots, \widehat{\eta}_t]^T$ denotes the vector formed by the first $t$ of those. This implies $\widehat{\mu}_t(x) = \alpha_t(x) + k_t(x)^T (K_t + \lambda I_t)^{-1} \widehat{N}_t$. Thus

$$k_t(x)^T (K_t + \lambda I_t)^{-1} \widehat{N}_t = \langle \varphi(x), \Phi_t^T (\Phi_t \Phi_t^T + \lambda I_t)^{-1} \widehat{N}_t \rangle_{\mathcal{H}} \stackrel{(a)}{=} \langle \varphi(x), \Phi_t^T \widehat{N}_t \rangle_{V_t^{-1}},$$

where $(a)$ uses equation 3. By Cauchy-Schwartz inequality, we have for any $x \in \mathcal{X}$

$$\left| k_t(x)^T (K_t + \lambda I_t)^{-1} \widehat{N}_t \right| \leq \|\varphi(x)\|_{V_t^{-1}} \left\| \Phi_t^T \widehat{N}_t \right\|_{V_t^{-1}} = \lambda^{-1/2} \left\| \Phi_t^T \widehat{N}_t \right\|_{V_t^{-1}} \sigma_t(x). \tag{8}$$

Now, by triangle inequality, we have

$$|f(x) - \widehat{\mu}_t(x)| \leq |f(x) - \alpha_t(x)| + \left| k_t(x)^T (K_t + \lambda I_t)^{-1} \widehat{N}_t \right|.$$

Hence from equation 7 and 8, we get

$$|f(x) - \widehat{\mu}_t(x)| \leq \left( B + \lambda^{-1/2} \left\| \Phi_t^T \widehat{N}_t \right\|_{V_t^{-1}} \right) \sigma_t(x). \tag{9}$$

Now, we define $\xi_t = \widehat{\eta}_t - \mathbb{E}\left[\widehat{\eta}_t | \mathcal{F}_{t-1}\right]$. Then, we have

$$\Phi_t^T \widehat{N}_t = \sum_{\tau=1}^{t} \widehat{\eta}_\tau \varphi(x_\tau) = \sum_{\tau=1}^{t} \xi_\tau \varphi(x_\tau) + \sum_{\tau=1}^{t} \mathbb{E}\left[\widehat{\eta}_\tau | \mathcal{F}_{\tau-1}\right] \varphi(x_\tau). \tag{10}$$

Observe that $\xi_t = \widehat{y}_t - \mathbb{E}\left[\widehat{y}_t | \mathcal{F}_{t-1}\right]$, and hence $|\xi_t| \leq 2b_t$. This implies that $\xi_t$ is zero-mean $2b_t$-sub-Gaussian random variable conditioned on $\mathcal{F}_{t-1}$. Further, observe that $\xi_t$ is $\mathcal{F}_t$- measurable and $x_t$ is $\mathcal{F}_{t-1}$- measurable. Hence, Lemma 7 implies that for any $\delta \in (0,1]$, with probability at least $1 - \delta$, for all $t \in \mathbb{N}$:

$$\left\| \sum_{\tau=1}^{t} \xi_\tau \varphi(x_\tau) \right\|_{V_t^{-1}} \leq 2b_t \sqrt{2\left(\frac{1}{2}\ln\left|I_\mathcal{H} + \lambda^{-1}\Phi_t^T\Phi_t\right| + \ln(1/\delta)\right)}$$

$$= 2b_t \sqrt{2\left(\frac{1}{2}\ln\left|I_t + \lambda^{-1}K_t\right| + \ln(1/\delta)\right)} \tag{11}$$

Now for any $a \in \mathbb{R}^t$,

$$\left\| \sum_{\tau=1}^{t} a_\tau \varphi(x_\tau) \right\|_{V_t^{-1}}^2 = \left\| \Phi_t^T a \right\|_{V_t^{-1}}^2 = a^T \Phi_t (\Phi_t^T \Phi_t + \lambda I_\mathcal{H})^{-1} \Phi_t^T a \overset{(a)}{=} a^T \Phi_t \Phi_t^T (\Phi_t \Phi_t^T + \lambda I_t)^{-1} a \overset{(b)}{\leq} \|a\|_2^2,$$

where $(a)$ follows from 3 and $(b)$ follows from the fact that $\Phi_t \Phi_t^T (\Phi_t \Phi_t^T + \lambda I_t)^{-1} \preceq I_t$. Therefore $\left\| \sum_{\tau=1}^{t} \mathbb{E}\left[\widehat{\eta}_\tau | \mathcal{F}_{\tau-1}\right] \varphi(x_\tau) \right\|_{V_t^{-1}}^2 \leq \sum_{\tau=1}^{t} \mathbb{E}\left[\widehat{\eta}_\tau | \mathcal{F}_{\tau-1}\right]^2$. Further, observe that $\mathbb{E}\left[\widehat{\eta}_t | \mathcal{F}_{t-1}\right] = \mathbb{E}\left[y_t \mathbb{1}_{|y_t| \leq b_t} | \mathcal{F}_{t-1}\right] - f(x_t) = -\mathbb{E}\left[y_t \mathbb{1}_{|y_t| > b_t} | \mathcal{F}_{t-1}\right]$. This implies

$$\left\| \sum_{\tau=1}^{t} \mathbb{E}\left[\widehat{\eta}_\tau | \mathcal{F}_{\tau-1}\right] \varphi(x_\tau) \right\|_{V_t^{-1}}^2 \leq \sum_{\tau=1}^{t} \mathbb{E}\left[y_\tau \mathbb{1}_{|y_\tau| > b_\tau} | \mathcal{F}_{\tau-1}\right]^2 \leq \sum_{\tau=1}^{t} \frac{1}{b_\tau^{2\alpha}} \mathbb{E}\left[|y_\tau|^{1+\alpha} | \mathcal{F}_{\tau-1}\right]^2 \leq v^2 \sum_{\tau=1}^{t} \frac{1}{b_\tau^{2\alpha}}.$$

Now setting $b_t = v^{\frac{1}{1+\alpha}} t^{\frac{1}{2(1+\alpha)}}$, we get

$$\left\| \sum_{\tau=1}^{t} \mathbb{E}\left[\widehat{\eta}_\tau | \mathcal{F}_{\tau-1}\right] \varphi(x_\tau) \right\|_{V_t^{-1}} \leq v^{\frac{1}{1+\alpha}} \sqrt{\sum_{\tau=1}^{t} \tau^{-\frac{\alpha}{1+\alpha}}} \leq v^{\frac{1}{1+\alpha}} \sqrt{\int_0^t \tau^{-\frac{\alpha}{1+\alpha}} d\tau} \leq \sqrt{2} v^{\frac{1}{1+\alpha}} t^{\frac{1}{2(1+\alpha)}}. \tag{12}$$

Combining 9, 10, 11 and 12, we have that for any $\delta \in (0,1]$, with probability at least $1 - \delta$, uniformly over all $t \geq 1$ and $x \in \mathcal{X}$:

$$|f(x) - \widehat{\mu}_t(x)| \leq \left( B + \sqrt{2/\lambda} \, v^{\frac{1}{1+\alpha}} t^{\frac{1}{2(1+\alpha)}} \left(1 + 2\sqrt{\frac{1}{2}\ln\left|I_t + \lambda^{-1}K_t\right| + \ln(1/\delta)}\right) \right) \sigma_t(x)$$

$$\leq \left( B + 3\sqrt{2/\lambda} \, v^{\frac{1}{1+\alpha}} t^{\frac{1}{2(1+\alpha)}} \sqrt{\frac{1}{2}\ln\left|I_t + \lambda^{-1}K_t\right| + \ln(1/\delta)} \right) \sigma_t(x). \tag{13}$$

Further observe that $|f(x) - \widehat{\mu}_0(x)| = |f(x)| = |\langle f, k(x, \cdot)\rangle_\mathcal{H}| \leq \|f\|_\mathcal{H} k^{1/2}(x, x) \leq B\sigma_0(x)$. Now the result follows by setting $\beta_{t+1} = B + \frac{3}{\sqrt{\lambda}} \, v^{\frac{1}{1+\alpha}} t^{\frac{1}{2(1+\alpha)}} \sqrt{\ln\left|I_t + \lambda^{-1}K_t\right| + 2\ln(1/\delta)}$, for all $t \geq 0$. ∎

Now, we will prove Theorem 1. For for any $\delta \in (0,1]$, we have, with probability at least $1 - \delta$, uniformly over all $t \geq 1$, the instantaneous regret of TGP-UCB (Algorithm 1) is

$$r_t = f(x^\star) - f(x_t)$$

$$\overset{(a)}{\leq} \widehat{\mu}_{t-1}(x^\star) + \beta_t \sigma_{t-1}(x^\star) - f(x_t)$$

$$\overset{(b)}{\leq} \widehat{\mu}_{t-1}(x_t) + \beta_t \sigma_{t-1}(x_t) - f(x_t)$$

$$\overset{(c)}{\leq} 2\beta_t \sigma_{t-1}(x_t).$$

Here $(a)$ and $(c)$ follow from 13, and $(b)$ is due to the choice of TGP-UCB(Algorithm 1). Since from Lemma 4, $\ln\left|I_t + \lambda^{-1}K_t\right| \leq \gamma_t$, we have $\beta_t \leq B + 3\sqrt{2/\lambda}\ v^{\frac{1}{1+\alpha}}t^{\frac{1}{2(1+\alpha)}}\sqrt{\gamma_t + \ln(1/\delta)}$, which is an increasing sequence $t$. Further, see that $\sum_{t=1}^{T}\sigma_{t-1}(x_t) \overset{(a)}{\leq} \sqrt{T\sum_{t=1}^{T}\sigma_{t-1}^2(x_t)} \overset{(b)}{\leq} \sqrt{2(1+\lambda)\gamma_T T}$, where $(a)$ is due to Cauchy-Schwartz inequality and $(b)$ is due to Lemma 6. Hence, for any $\delta \in (0,1]$, with probability at least $1-\delta$, the cumulative regret of TGP-UCB after $T$ rounds is

$$R_T = O\left(B\sqrt{T\gamma_T} + v^{\frac{1}{1+\alpha}}\sqrt{\gamma_T(\gamma_T + \ln(1/\delta))}T^{\frac{2+\alpha}{2(1+\alpha)}}\right).$$

## C   Regret lower bound: proof of Theorem 2

Our analysis builds heavily on that of the optimization setting with $f \in \mathcal{H}_k(\mathcal{X})$ and with Gaussian noise studied in [31], but with important differences. Roughly speaking, we use the same construction of $f$ as in [31], but we construct the rewards differently to capture the heavy-tailed scenario. We now proceed with the formal proof.

### C.1   Construction of the ground-truth function

- Let $g(x)$ be a function on $\mathbb{R}^d$ with the following properties:

  1. The RKHS norm of $g$ is bounded: $\|g\|_{\mathcal{H}} \leq B$.

  2. $|g(x)| \leq 2\Delta$ with a maximum value of $2\Delta$ at $x = 0$ and $g(x) < \Delta$ when $\|x\|_\infty > w$ for some $w > 0$ and $\Delta > 0$, to be chosen later.

- Letting $g(x)$ be such a function, we construct $M$ functions $f_1, \ldots, f_M$ first by shifting $g$ such that each $f_j$ has its maximum at a unique point in a uniform grid, and then by restricting them to the domain $\mathcal{X} = [0,1]^d$. Using a step size $w$ in each dimension, one can construct a grid of size $M = \lfloor\left(\frac{1}{w}\right)^d\rfloor$ of the domain $\mathcal{X}$, and hence $M$ such functions $f_j$. In this process we ensure that any $\Delta$-optimal point for $f_j$ fails to be $\Delta$-optimal point for any other $f_{j'}$.

- Finally, we choose $f$ as a uniformly sampled function from the set $\{f_1, \ldots, f_M\}$.

It remains to choose $g$, $w$, and $\Delta$ so that the above properties are satisfied.

- For some absolute constant $\zeta > 0$ we choose $g(x) = \frac{2\Delta}{h(0)}h(\frac{x\zeta}{w})$, where $h$ is the inverse Fourier transform of the *multi-dimensional bump function*: $H(\omega) = e^{-\frac{1}{1-\|\omega\|_2^2}}\mathbb{1}_{\{\|\omega\|_2^2 \leq 1\}}$. Note that since $H$ is real and symmetric, the maximum of $h$ is attained at $x = 0$, and hence the maximum of $g$ is $g(0) = 2\Delta$, as desired. Further, since $H$ has finite energy, $h(x) \to 0$ as $\|x\|_2 \to \infty$. Hence, there exists an absolute constant $\zeta$ such that $h(x) < \frac{1}{2}h(0)$ when $\|x\|_\infty > \zeta$, and thus $g(x) < \Delta$ for $\|x\|_\infty > w$, as desired.

- It now remains to choose $w$ and $\Delta$ to ensure that $\|g\|_{\mathcal{H}} \leq B$, for a given $B$. Note that, while a smaller $\Delta$ ensures a low RKHS norm, a smaller $w$ increases it. Hence, as long as $\Delta$ is very small, we can afford to take $w << 1$, so that there is no risk of having $M = 0$. For $\frac{\Delta}{B} << 1$, it is shown in [31] that the condition $\|g\|_{\mathcal{H}} \leq B$ can be achieved with $w = \frac{\zeta\pi l}{\sqrt{\ln\frac{B(2\pi l^2)^{d/4}h(0)}{2\Delta}}}$ for the SE kernel, and with $w = \zeta\left(\frac{2\Delta(8\pi^2)^{(\nu+d/2)/2}}{Bc^{-1/2}h(0)}\right)^{1/\nu}$ for the Matérn kernel for some $c > 0$. We consider $\Delta$ as arbitrary for now, but later this will be chosen to ensure that $\frac{\Delta}{B}$ is sufficiently small.

- From the choice of $w$, we see that $M = \Theta\left((\ln\frac{B}{\Delta})^{\frac{d}{2}}\right)$ for the SE kernel, and $M = \Theta\left((\frac{B}{\Delta})^{\frac{d}{\nu}}\right)$ for the Matérn kernel. Note that the assumption of sufficiently small $\frac{\Delta}{B}$ in ensures that $M >> 1$, i.e. there are enough number of functions to sample from.

## C.2 Construction of the reward distribution

For any given $\alpha \in (0, 1]$, $v > 0$ and $x \in [0, 1]^d$, we define the reward distribution as

$$y(x) = \begin{cases} sgn\left(f(x)\right) \left(\frac{v}{2\Delta}\right)^{\frac{1}{\alpha}} & \text{with probability } \left(\frac{2\Delta}{v}\right)^{\frac{1}{\alpha}} |f(x)|, \\ 0 & \text{otherwise.} \end{cases} \tag{14}$$

Note that 14 is a valid probability distribution as long as $\Delta \leq \frac{1}{2} v^{\frac{1}{1+\alpha}}$. Then, $\mathbb{E}\left[y(x)\right] = |f(x)| \, sgn((f(x)) = f(x)$ and $\mathbb{E}\left[|y(x)|^{1+\alpha}\right] = \left(\frac{v}{2\Delta}\right)^{\frac{1+\alpha}{\alpha}} \left(\frac{2\Delta}{v}\right)^{\frac{1}{\alpha}} |f(x)| = \frac{v|f(x)|}{2\Delta} \leq v$ for any $\alpha \in (0, 1]$. Thus, we ensure that the $(1 + \alpha)$-th absolute moment of the rewards are upper bounded by $v$.

## C.3 Preliminary notations and lemmas

Now, we introduce the following notations, also used in [31]:

- $y_m$ denote the reward function when the underlying ground truth is $f_m$ for $m = 1, \dots, M$. $f_0$ denotes the function which is zero everywhere, and $y_0$ the corresponding reward function. $P_m(Y_T)$ (resp. $P_0(Y_T)$) denotes the probability density function of the reward sequence $Y_T = \{y_1, \dots, y_T\}$ when the underlying function is $f_m$ (resp. $f_0$). $P_m(y|x)$ (resp. $P_0(y|x)$) denotes the conditional density of the reward $y$ given the selected point $x$ when the underlying function is $f_m$ (resp. $f_0$).

- $\mathbb{E}_m$ (resp. $\mathbb{E}_0$) and $\mathbb{P}_m$ (resp. $\mathbb{P}_0$) denote expectations and probabilities (with respect to the noisy rewards) when the underlying function is $f_m$ (resp. $f_0$). $\mathbb{E}[\cdot] = \frac{1}{M} \sum_{m=1}^{M} \mathbb{E}_m[\cdot]$ (resp. $\mathbb{P}_m[\cdot]$) denote the expectation (resp. probability) with respect to the noisy rewards and $f$ drawn uniformly from $\{f_1, \dots, f_M\}$.

- $\{\mathcal{R}_m\}_{m=1}^{M}$ denote a partition of $\mathcal{X}$ into $M$ regions such that each $f_m, m = 1, \dots, M$ has its maximum at the center of $\mathcal{R}_m$. $v_m^j = \max_{x \in \mathcal{R}_j} |f_m(x)|$ denotes the maximum absolute value of $f_m$ in the region $\mathcal{R}_j$ and $D_m^j = \max_{x \in \mathcal{R}_j} D_{\text{KL}}\left(P_0(\cdot|x)||P_m(\cdot|x)\right)$ denotes the maximum KL divergence between $P_0(\cdot|x)$ and $P_m(\cdot|x)$ within $\mathcal{R}_j$. $N_j = \sum_{t=1}^{T} \mathbb{1}_{\{x_t \in \mathcal{R}_j\}}$ denotes the number of points within $\mathcal{R}_j$ that are selected up to time $T$.

Next, we present some useful lemmas from [31].

**Lemma 9** *[31, Lemma 3] Under the preceding definitions, we have* $\mathbb{E}_m[N_j] \leq \mathbb{E}_0[N_j] + T\sqrt{D_{KL}(P_0||P_m)}$ *for all* $m = 1, \dots, M$ *and* $j = 1, \dots, M$.

**Lemma 10** *[31, Lemma 4] Under the preceding definitions, we have* $D_{KL}(P_0||P_m) \leq \sum_{j=1}^{M} \mathbb{E}_0[N_j]D_m^j$ *for all* $m = 1, \dots, M$.

**Lemma 11** *[31, Lemma 5] The functions $f_m$ constructed in Section C.1 are such that the quantities $v_m^j$ satisfy:*
*(a)* $\sum_{m=1}^{M} v_m^j = O(\Delta)$ *for all* $j = 1, \dots, M$ *and (b)* $\sum_{j=1}^{M} v_m^j = O(\Delta)$ *for all* $m = 1, \dots, M$.

## C.4 Analysis of expected cumulative regret

Observe that $\mathbb{E}_m[f(x_t)] \leq \sum_{j=1}^{M} \mathbb{P}_m[x_t \in \mathcal{R}_j]v_m^j$. This implies

$$\mathbb{E}_m\left[\sum_{t=1}^{T} f(x_t)\right] \leq \sum_{j=1}^{M} v_m^j \mathbb{E}_m[N_j] \leq \sum_{j=1}^{M} v_m^j \left(\mathbb{E}_0[N_j] + T\sqrt{\sum_{j'=1}^{M} \mathbb{E}_0[N_{j'}]D_m^{j'}}\right),$$

where the last inequality follows from Lemma 9. Now averaging over $m = 1, \dots, M$ we obtain the following:

$$\mathbb{E}\left[\sum_{t=1}^{T} f(x_t)\right] \leq \frac{1}{M} \sum_{m=1}^{M} \sum_{j=1}^{M} v_m^j \left(\mathbb{E}_0[N_j] + T\sqrt{\sum_{j'=1}^{m} \mathbb{E}_0[N_{j'}]D_m^{j'}}\right). \tag{15}$$

We can bound the first term as follows:

$$\frac{1}{M}\sum_{m=1}^{M}\sum_{j=1}^{M}v_m^j\mathbb{E}_0[N_j] = \frac{1}{M}\sum_{j=1}^{M}\sum_{m=1}^{M}v_m^j\mathbb{E}_0[N_j] \stackrel{(a)}{=} O\left(\frac{\Delta}{M}\right)\sum_{j=1}^{M}\mathbb{E}_0[N_j] \stackrel{(b)}{=} O\left(\frac{T\Delta}{M}\right), \quad (16)$$

where $(a)$ follows from part $(a)$ of Lemma 11, and $(b)$ follows from $\sum_{j=1}^{M}N_j = T$. In order to bound the second term, first we note that $y_0(x) = 0$ for all $x \in \mathcal{X}$. Therefore, we have

$$D_{\mathrm{KL}}\left(P_0(\cdot|x)||P_m(\cdot|x)\right) = \ln\frac{1}{1-\left(\frac{2\Delta}{v}\right)^{\frac{1}{\alpha}}|f_m(x)|} \stackrel{(a)}{\leq} \frac{\left(\frac{2\Delta}{v}\right)^{\frac{1}{\alpha}}|f_m(x)|}{1-\left(\frac{2\Delta}{v}\right)^{\frac{1}{\alpha}}|f_m(x)|}$$

$$\stackrel{(b)}{\leq} \frac{\left(\frac{2\Delta}{v}\right)^{\frac{1}{\alpha}}|f_m(x)|}{1-(2\Delta)^{\frac{1+\alpha}{\alpha}}v^{-\frac{1}{\alpha}}}$$

$$\stackrel{(c)}{\leq} 2\left(\frac{2\Delta}{v}\right)^{\frac{1}{\alpha}}|f_m(x)|.$$

Here $(a)$ holds because $\ln(x) \leq x-1$ for all $x \geq 1$, $(b)$ holds as $|f(x)| \leq 2\Delta$ and $(c)$ holds for $\Delta \leq \frac{1}{2}\left(\frac{1}{2}\right)^{\frac{\alpha}{1+\alpha}}v^{\frac{1}{1+\alpha}}$. Observe that this choice of $\Delta$ is compatible with 14. This implies that for all $j = 1,\ldots,M$,

$$D_m^j \leq 2^{\frac{1+\alpha}{\alpha}}\left(\frac{\Delta}{v}\right)^{\frac{1}{\alpha}}v_m^j \text{ if } \Delta \leq \frac{1}{2}\left(\frac{1}{2}\right)^{\frac{\alpha}{1+\alpha}}v^{\frac{1}{1+\alpha}}. \quad (17)$$

Now, we can bound the second term as follows:

$$\frac{1}{M}\sum_{m=1}^{M}\sum_{j=1}^{M}v_m^j\sqrt{\sum_{j'=1}^{m}\mathbb{E}_0[N_{j'}]D_m^{j'}} \stackrel{(a)}{=} O(\Delta)\frac{1}{M}\sum_{m=1}^{M}\sqrt{\sum_{j'=1}^{M}\mathbb{E}_0[N_{j'}]D_m^{j'}}$$

$$\stackrel{(b)}{\leq} O(\Delta)\sqrt{\frac{1}{M}\sum_{m=1}^{M}\sum_{j'=1}^{M}\mathbb{E}_0[N_{j'}]D_m^{j'}}$$

$$\stackrel{(c)}{\leq} O(\Delta)2^{\frac{1+\alpha}{2\alpha}}\left(\frac{\Delta}{v}\right)^{\frac{1}{2\alpha}}\sqrt{\frac{1}{M}\sum_{m=1}^{M}\sum_{j'=1}^{M}\mathbb{E}_0[N_{j'}]v_m^{j'}}$$

$$\stackrel{(d)}{=} O(\Delta)2^{\frac{1+\alpha}{2\alpha}}\left(\frac{\Delta}{v}\right)^{\frac{1}{2\alpha}}\sqrt{O\left(\frac{\Delta}{M}\right)\sum_{j'=1}^{M}\mathbb{E}_0[N_{j'}]}$$

$$\stackrel{(e)}{=} O\left(\Delta\frac{(2\Delta)^{\frac{1+\alpha}{2\alpha}}}{v^{\frac{1}{2\alpha}}}\sqrt{\frac{T}{M}}\right). \quad (18)$$

Here $(a)$ follows from part $(b)$ of Lemma 11, $(b)$ follows from Jensen's inequality, $(c)$ follows from 17 if $\Delta \leq \frac{1}{2}\left(\frac{1}{2}\right)^{\frac{\alpha}{1+\alpha}}v^{\frac{1}{1+\alpha}}$, $(d)$ follows from part $(a)$ of Lemma 11, and $(e)$ follows from $\sum_{j=1}^{M}N_j = T$. Substituting 16 and 18 in 15 gives

$$\mathbb{E}\left[\sum_{t=1}^{T}f(x_t)\right] \leq CT\Delta\left(\frac{1}{M} + \frac{(2\Delta)^{\frac{1+\alpha}{2\alpha}}}{v^{\frac{1}{2\alpha}}}\sqrt{\frac{T}{M}}\right) \text{ for } \Delta \leq \frac{1}{2}\left(\frac{1}{2}\right)^{\frac{\alpha}{1+\alpha}}v^{\frac{1}{1+\alpha}}. \quad (19)$$

Since $f(x^\star) = 2\Delta$, the expected cumulative regret

$$\mathbb{E}[R_T] = Tf(x^\star) - \mathbb{E}\left[\sum_{t=1}^{T}f(x_t)\right] \geq T\Delta\left(2 - \frac{C}{M} - \frac{C(2\Delta)^{\frac{1+\alpha}{2\alpha}}}{v^{\frac{1}{2\alpha}}}\sqrt{\frac{T}{M}}\right) \text{ for } \Delta \leq \frac{1}{2}\left(\frac{1}{2}\right)^{\frac{\alpha}{1+\alpha}}v^{\frac{1}{1+\alpha}}.$$

Since $M \to \infty$ as $\frac{\Delta}{B} \to 0$, we have $\frac{C}{M} \leq \frac{1}{2}$ for sufficiently small $\frac{\Delta}{B}$. Hence, we have

$$\mathbb{E}[R_T] \geq T\Delta\left(\frac{3}{2} - C\frac{(2\Delta)^{\frac{1+\alpha}{2\alpha}}}{v^{\frac{1}{2\alpha}}}\sqrt{\frac{T}{M}}\right)$$

$$\geq T\Delta \quad \text{for } \Delta \leq \frac{1}{2}\left(\min\left\{\frac{1}{2},\frac{M}{4C^2T}\right\}\right)^{\frac{\alpha}{1+\alpha}}v^{\frac{1}{1+\alpha}}. \quad (20)$$

Now, if $M \leq 2C^2T$, then

$$\mathbb{E}\left[R_T\right] = \Omega\left(v^{\frac{1}{1+\alpha}} M^{\frac{\alpha}{1+\alpha}} T^{\frac{1}{1+\alpha}}\right) \text{ for } \frac{1}{4}\left(\frac{M}{4C^2T}\right)^{\frac{\alpha}{1+\alpha}} v^{\frac{1}{1+\alpha}} \leq \Delta \leq \frac{1}{2}\left(\frac{M}{4C^2T}\right)^{\frac{\alpha}{1+\alpha}} v^{\frac{1}{1+\alpha}}. \quad (21)$$

### C.4.1 Application to the squared exponential kernel

For the SE kernel, we have from the choice $M = \Theta\left(\left(\ln \frac{B}{\Delta}\right)^{\frac{d}{2}}\right)$, along with the upper and lower bounds on $\Delta$ in 21, that $\Delta = \Theta\left(\left(\frac{1}{T}\left(\ln \frac{B}{\Delta}\right)^{\frac{d}{2}}\right)^{\frac{\alpha}{1+\alpha}} v^{\frac{1}{1+\alpha}}\right)$. This, in turn, implies that $\ln \frac{B}{\Delta} = \ln \frac{BT^{\frac{\alpha}{1+\alpha}}}{v^{\frac{1}{1+\alpha}}} - \ln\left(\Theta(1)\left(\ln \frac{B}{\Delta}\right)^{\frac{d\alpha}{2(1+\alpha)}}\right)$. Since $d = O(1)$ and $\frac{\alpha}{1+\alpha} \in (0, \frac{1}{2}]$, the second term behaves as $\Theta(\ln \ln \frac{B}{\Delta})$, which is $\Theta\left(\frac{1}{2}\ln \frac{B}{\Delta}\right)$ for sufficiently small $\frac{\Delta}{B}$. This, implies that $\ln \frac{B}{\Delta} = \Theta\left(\ln \frac{BT^{\frac{\alpha}{1+\alpha}}}{v^{\frac{1}{1+\alpha}}}\right)$, and thus, in turn, $M = \Theta\left(\left(\ln \frac{BT^{\frac{\alpha}{1+\alpha}}}{v^{\frac{1}{1+\alpha}}}\right)^{\frac{d}{2}}\right)$ and $\Delta = \Theta\left(v^{\frac{1}{1+\alpha}}\left(\ln \frac{BT^{\frac{\alpha}{1+\alpha}}}{v^{\frac{1}{1+\alpha}}}\right)^{\frac{d\alpha}{2(1+\alpha)}} T^{-\frac{\alpha}{1+\alpha}}\right)$.
Note that the choice of $M$ ensures that $M \leq 2C^2T$ and the choice of $\Delta$ ensures that $\frac{\Delta}{B}$ is indeed sufficiently small as long as $v^{\frac{1}{1+\alpha}} \leq C'BT^{\frac{\alpha}{1+\alpha}}$ for some sufficiently small constant $C'$ [5]. Now, substituting $M$ in 21, we obtain $\mathbb{E}\left[R_T\right] = \Omega\left(v^{\frac{1}{1+\alpha}}\left(\ln \frac{BT^{\frac{\alpha}{1+\alpha}}}{v^{\frac{1}{1+\alpha}}}\right)^{\frac{d\alpha}{2(1+\alpha)}} T^{\frac{1}{1+\alpha}}\right) = \Omega\left(v^{\frac{1}{1+\alpha}}\left(\ln \frac{B^{\frac{1+\alpha}{\alpha}}T}{v^{\frac{1}{\alpha}}}\right)^{\frac{d\alpha}{2(1+\alpha)}} T^{\frac{1}{1+\alpha}}\right)$, since, generally, $d = O(1)$ and $\frac{\alpha}{1+\alpha} \in (0, \frac{1}{2}]$.

### C.4.2 Application to the Matérn kernel

For the Matérn kernel, we have from the choice $M = \Theta\left(\left(\frac{B}{\Delta}\right)^{\frac{d}{\nu}}\right)$, along with the upper and lower bounds on $\Delta$ in 21, that $\Delta = \Theta\left(\left(\frac{1}{T}\left(\frac{B}{\Delta}\right)^{\frac{d}{\nu}}\right)^{\frac{\alpha}{1+\alpha}} v^{\frac{1}{1+\alpha}}\right)$. This, in turn, implies that $\Delta = \Theta\left(v^{\frac{\nu/(1+\alpha)}{\nu+d\alpha/(1+\alpha)}} B^{\frac{d\alpha/(1+\alpha)}{\nu+d\alpha/(1+\alpha)}} T^{-\frac{\nu\alpha/(1+\alpha)}{\nu+d\alpha/(1+\alpha)}}\right)$ and $M = \Theta\left(v^{-\frac{d/(1+\alpha)}{\nu+d\alpha/(1+\alpha)}} B^{\frac{d}{\nu+d\alpha/(1+\alpha)}} T^{\frac{d\alpha/(1+\alpha)}{\nu+d\alpha/(1+\alpha)}}\right)$.
Once again, we see that the choice of $M$ ensures that $M \leq 2C^2T$ and the choice of $\Delta$ ensures that $\frac{\Delta}{B}$ is indeed sufficiently small as long as $v^{\frac{1}{1+\alpha}} \leq C'BT^{\frac{\alpha}{1+\alpha}}$ for some sufficiently small constant $C'$.
Now, substituting $M$ in 21, we obtain $\mathbb{E}\left[R_T\right] = \Omega\left(v^{\frac{\nu/(1+\alpha)}{\nu+d\alpha/(1+\alpha)}} B^{\frac{d\alpha/(1+\alpha)}{\nu+d\alpha/(1+\alpha)}} T^{\frac{1}{1+\alpha}\frac{\nu+d\alpha}{\nu+d\alpha/(1+\alpha)}}\right) = \Omega\left(v^{\frac{\nu}{\nu(1+\alpha)+d\alpha}} B^{\frac{d\alpha}{\nu(1+\alpha)+d\alpha}} T^{\frac{\nu+d\alpha}{\nu(1+\alpha)+d\alpha}}\right)$.

## D Analysis of ATA-GP-UCB

### D.1 Construction of tighter confidence set using data adaptive truncation

The following lemma helps us to show that $(1+\alpha)$-th norm of $u_i \in \mathbb{R}^t$ is $t^{\frac{1-\alpha}{2(1+\alpha)}}$, where $u_i^T, i \in [m_t]$ are the rows of $\tilde{V}_t^{-1/2}\tilde{\Phi}_t^T$.

**Lemma 12** *Let $A \in \mathbb{R}^{p \times q}$. Let $c_i \in \mathbb{R}^p, i = 1, \ldots, q$ be the $i$-th column of $A(A^TA + \lambda I_q)^{-1/2}$. Then for any $\beta \in [1, \infty)$, we have $\|c_i\|_\beta \leq p^{\frac{2-\beta}{2\beta}}$ for all $i \in [q]$.*

**Proof** Let the singular value decomposition of $A$ be $U\Sigma V^T$, where $U$ and $V$ are unitary matrices. This implies $A(A^TA + \lambda I_q)^{-1/2} = U\Sigma(\Sigma^T\Sigma + \lambda I_q)^{-1/2}V^T$. Now, the $i$-th column of $A(A^TA +$

$\lambda I_q)^{-1/2}$ is given by $c_i = U\Sigma(\Sigma^T\Sigma + \lambda I)^{-1/2}V^T e_i$. Therefore,

$$\|c_i\|_2 = \left\|U\Sigma(\Sigma^T\Sigma + \lambda I)^{-1/2}V^T e_i\right\|_2 = \left\|\Sigma(\Sigma^T\Sigma + \lambda I)^{-1/2}V^T e_i\right\|_2$$
$$\leq \left\|\Sigma(\Sigma^T\Sigma + \lambda I)^{-1/2}\right\|_2 \|V^T e_i\|_2 \leq 1.$$

Now the result follows from the fact that for any $a \in \mathbb{R}^p, \|a\|_2 \leq 1$ the maximum value of $\|a\|_\beta$ for any $\beta \in [1, \infty)$ is $p^{\frac{2-\beta}{2\beta}}$ with the maximum attained at $[\frac{1}{\sqrt{p}}, \ldots, \frac{1}{\sqrt{p}}]^T$. ∎

Now, we will show that the data adaptive truncation of ATA-GP-UCB helps us to achieve tighter confidence sets than TGP-UCB.

**Lemma 13 (Effect of data adaptive truncation)** *For any $\delta \in (0, 1]$, ATA-GP-UCB with $b_t = (v/\ln(2m_tT/\delta))^{\frac{1}{1+\alpha}} t^{\frac{1-\alpha}{2(1+\alpha)}}$, ensures, with probability at least $1 - \delta$, that uniformly over all $t \in [T]$,*

$$\left\|\tilde{V}_t^{-1}\tilde{\Phi}_t^T f_t - \tilde{\theta}_t\right\|_{\tilde{V}_t} \leq 4\sqrt{m_t}\, v^{\frac{1}{1+\alpha}}\, (\ln(2m_tT/\delta))^{\frac{\alpha}{1+\alpha}}\, t^{\frac{1-\alpha}{2(1+\alpha)}},$$

*where $f_t = [f(x_1), \ldots, f(x_t)]^T$ is a vector containing $f$'s evaluations up to round $t$.*

**Proof** The proof is inspired from Shao et al. [34], with some changes. Fix any $t \in \mathbb{N}$. Let $u_i^T \in \mathbb{R}^{1\times t}$, $i = 1, \ldots, m_t$ denotes the $i$-th row of $\tilde{V}_t^{-1/2}\tilde{\Phi}_t^T$ where $\tilde{V}_t = \tilde{\Phi}_t^T\tilde{\Phi}_t + \lambda I_{m_t}$. Let $r_i = u_i^T Y_t = \sum_{\tau=1}^t u_{i,\tau} y_\tau$ denotes the sum of weighted historical rewards in the $i$-th dimension of the feature space with the weight vector $u_i$ and $\hat{r}_i = \sum_{\tau=1}^t u_{i,\tau} y_\tau \mathbb{1}_{|u_{i,\tau} y_\tau| \leq b_t}$ denotes the corresponding truncation. Let $\mathcal{F}'_{t,\tau} = \sigma(\{x_1, \ldots, x_t\} \cup \{y_1, \ldots, y_\tau\}), \tau = 0, 1, 2, \ldots, t$ denotes the $\sigma$-algebra generated by the arms played up to time $t$ and rewards obtained up to time $\tau$. Observe that $\mathcal{F}'_{t,0} \subseteq \mathcal{F}'_{t,1} \subseteq \mathcal{F}'_{t,2} \subseteq \ldots$ and define $\mathcal{F}'_t = \mathcal{F}'_{t,0}$. Then, $\mathbb{E}[Y_t|\mathcal{F}'_t] = f_t$ and $u_i, i = 1, \ldots, m_t$ are $\mathcal{F}'_t$-measurable. Therefore, we have $\mathbb{E}[r_i|\mathcal{F}'_t] = u_i^T f_t = \sum_{\tau=1}^t u_{i,\tau} f(x_\tau) = \sum_{\tau=1}^t \mathbb{E}[u_{i,\tau} y_\tau|\mathcal{F}'_{t,\tau-1}]$ for all $i \in [m_t]$. This implies

$$|\hat{r}_i - \mathbb{E}[r_i|\mathcal{F}'_t]|$$
$$= \left|\sum_{\tau=1}^t u_{i,\tau} y_\tau \mathbb{1}_{|u_{i,\tau} y_\tau| \leq b_t} - \sum_{\tau=1}^t \mathbb{E}[u_{i,\tau} y_\tau|\mathcal{F}'_{t,\tau-1}]\right|$$
$$= \left|\sum_{\tau=1}^t u_{i,\tau} y_\tau \mathbb{1}_{|u_{i,\tau} y_\tau| \leq b_t} - \sum_{\tau=1}^t \mathbb{E}\left[u_{i,\tau} y_\tau \left(\mathbb{1}_{|u_{i,\tau} y_\tau| \leq b_t} + \mathbb{1}_{|u_{i,\tau} y_\tau| > b_t}\right)|\mathcal{F}'_{t,\tau-1}\right]\right|$$
$$\leq \left|\sum_{\tau=1}^t \left(u_{i,\tau} y_\tau \mathbb{1}_{|u_{i,\tau} y_\tau| \leq b_t} - \mathbb{E}[u_{i,\tau} y_\tau \mathbb{1}_{|u_{i,\tau} y_\tau| \leq b_t}|\mathcal{F}'_{t,\tau-1}]\right)\right| + \sum_{\tau=1}^t \mathbb{E}[|u_{i,\tau} y_\tau|\mathbb{1}_{|u_{i,\tau} y_\tau| > b_t}|\mathcal{F}'_{t,\tau-1}].$$

Now, we will bound the second term first. Observe that $\mathbb{E}[|u_{i,\tau} y_\tau|\mathbb{1}_{|u_{i,\tau} y_\tau| > b_t}|\mathcal{F}'_{t,\tau-1}] \leq b_t^{-\alpha}\mathbb{E}[|u_{i,\tau} y_\tau|^{1+\alpha}\mathbb{1}_{|u_{i,\tau} y_\tau| > b_t}|\mathcal{F}'_{t,\tau-1}] \leq b_t^{-\alpha}|u_{i,\tau}|^{1+\alpha}\mathbb{E}[|y_\tau|^{1+\alpha}|\mathcal{F}'_{t,\tau-1}]$. Now since the noise variables are sampled independent of the arms played, it holds that $\mathbb{E}[|y_\tau|^{1+\alpha}|\mathcal{F}'_{t,\tau-1}] = \mathbb{E}[|y_\tau|^{1+\alpha}|\mathcal{F}_{\tau-1}]$ and therefore

$$\sum_{\tau=1}^t \mathbb{E}[|u_{i,\tau} y_\tau|\mathbb{1}_{|u_{i,\tau} y_\tau| > b_t}|\mathcal{F}'_{t,\tau-1}] \leq v b_t^{-\alpha}\sum_{\tau=1}^t |u_{i,\tau}|^{1+\alpha}.$$

Now, we will bound the first term. For that, we define $M_{t,\tau} := u_{i,\tau} y_\tau \mathbb{1}_{|u_{i,\tau} y_\tau| \leq b_t} - \mathbb{E}[u_{i,\tau} y_\tau \mathbb{1}_{|u_{i,\tau} y_\tau| \leq b_t} | \mathcal{F}'_{t,\tau-1}], \tau = 1, 2, \ldots, t$. It is easy to see that $(M_{t,\tau})_{\tau \geq 1}$ is a martingale difference sequence with respect to the filtration $(\mathcal{F}'_{t,\tau})_{\tau \geq 0}$ and $|M_{t,\tau}| \leq 2b_t$ almost surely. Further, $\mathbb{V}[M_\tau \mid \mathcal{F}'_{t,\tau-1}] = \mathbb{V}[u_{i,\tau} y_\tau \mathbb{1}_{|u_{i,\tau} y_\tau| \leq b_t} \mid \mathcal{F}'_{t,\tau-1}] \leq \mathbb{E}[u_{i,\tau}^2 y_\tau^2 \mathbb{1}_{|u_{i,\tau} y_\tau| \leq b_t} \mid \mathcal{F}'_{t,\tau-1}] \leq b_t^{1-\alpha}|u_{i,\tau}|^{1+\alpha}\mathbb{E}[|y_\tau|^{1+\alpha} \mid \mathcal{F}'_{t,\tau-1}] \leq v b_t^{1-\alpha}|u_{i,\tau}|^{1+\alpha}$. Then by Bernstein's inequality [32], we have that for any $\gamma \in [0, 1/2b_t]$ and $\delta \in (0, 1]$, with probability at least $1 - \delta$,

$$\left|\sum_{\tau=1}^t \left(u_{i,\tau} y_\tau \mathbb{1}_{|u_{i,\tau} y_\tau| \leq b_t} - \mathbb{E}[u_{i,\tau} y_\tau \mathbb{1}_{|u_{i,\tau} y_\tau| \leq b_t}]\right)\right| \leq \frac{1}{\gamma}\ln(2/\delta) + \gamma(e - 2)\sum_{\tau=1}^t v b_t^{1-\alpha}|u_{i,\tau}|^{1+\alpha}.$$

Now setting $\gamma = 1/2b_t$, we obtain that for any $i \in [m_t]$ and $\delta \in (0, 1]$, with probability at least $1 - \delta$,

$$
\begin{aligned}
|\widehat{r}_i - \mathbb{E}[r_i|\mathcal{F}'_t]| &\leq 2b_t \ln(2/\delta) + 2vb_t^{-\alpha} \sum_{\tau=1}^{t} |u_{i,\tau}|^{1+\alpha} \\
&= 2b_t \ln(2/\delta) + 2vb_t^{-\alpha} \|u_i\|_{1+\alpha}^{1+\alpha} \\
&\overset{(a)}{\leq} 2b_t \ln(2/\delta) + 2vb_t^{-\alpha} t^{\frac{1-\alpha}{2}} \\
&\overset{(b)}{\leq} 4v^{\frac{1}{1+\alpha}} (\ln(2/\delta))^{\frac{\alpha}{1+\alpha}} t^{\frac{1-\alpha}{2(1+\alpha)}}.
\end{aligned}
\tag{22}
$$

Here $(a)$ follows from Lemma 12 and $(b)$ holds for $b_t = (v/\ln(2/\delta))^{\frac{1}{1+\alpha}} t^{\frac{1-\alpha}{2(1+\alpha)}}$. Now observe that $\tilde{V}_t^{1/2}\tilde{\theta}_t = [\widehat{r}_1, \dots, \widehat{r}_{m_t}]^T$ and $\tilde{V}_t^{-1/2}\tilde{\Phi}_t^T f_t = [u_1^T f_t, \dots, u_{m_t}^T f_t]^T = [\mathbb{E}[r_1|\mathcal{F}'_t], \dots, \mathbb{E}[r_{m_t}|\mathcal{F}'_t]]^T$. This implies

$$
\left\| \tilde{V}_t^{-1} \tilde{\Phi}_t^T f_t - \tilde{\theta}_t \right\|_{\tilde{V}_t} = \left\| \tilde{V}_t^{-1/2} \tilde{\Phi}_t^T f_t - \tilde{V}_t^{1/2} \tilde{\theta}_t \right\|_2 = \sqrt{\sum_{i=1}^{m_t} \left( \widehat{r}_i - \mathbb{E}[r_i|\mathcal{F}'_{t-1}] \right)^2}.
$$

Therefore, by taking an union bound over all $i \in [m_t]$ and setting $\delta = \delta/m_t$ in 22, we obtain that for any $t \in \mathbb{N}$ and $\delta \in (0, 1]$, with probability at least $1 - \delta$,

$$
\left\| \tilde{V}_t^{-1} \tilde{\Phi}_t^T f_t - \tilde{\theta}_t \right\|_{\tilde{V}_t} \leq 4\sqrt{m_t} \, v^{\frac{1}{1+\alpha}} (\ln(2m_t/\delta))^{\frac{\alpha}{1+\alpha}} t^{\frac{1-\alpha}{2(1+\alpha)}}.
$$

Now the result follows by taking another union bound over all $t \in [T]$ and setting $\delta = \delta/T$. ∎

### D.2 Analysis of ATA-GP-UCB under quadrature Fourier features (QFF) approximation

#### D.2.1 Error due to Fourier feature approximation

**Definition 2 (Uniform Approximation [26])** *Let* $k : \mathcal{X} \times \mathcal{X} \to \mathbb{R}, \mathcal{X} \subset \mathbb{R}^d$ *be a kernel, then a feature map* $\tilde{\varphi} : \mathcal{X} \to \mathbb{R}^m$ *uniformly approximates* $k$ *within an accuracy* $\varepsilon_m$ *if and only if,*

$$
\sup_{x,y \in \mathcal{X}} \left| k(x,y) - \tilde{\varphi}(x)^T \tilde{\varphi}(y) \right| \leq \varepsilon_m.
\tag{23}
$$

**Lemma 14 (QFF error)** *[26, Theorem 1] Let* $\mathcal{X} = [0,1]^d$, $k = k_{SE}$ *and* $\tilde{\varphi}$ *be as in 1. Then,*

$$
\varepsilon_m \leq d2^{d-1} \frac{1}{\sqrt{2}\bar{m}^{\bar{m}}} \left( \frac{e}{4l^2} \right)^{\bar{m}}.
$$

Lemma 14 implies that QFF embedding (1) of $k_{\text{SE}}$ satisfies $\varepsilon_m = O\left( \frac{d2^{d-1}}{(\bar{m}l^2)^{\bar{m}}} \right)$ where $m = \bar{m}^d$. We can achieve exponential decay only when $\bar{m} > 1/l^2$, and in that case $O\left((d + \ln(d/\varepsilon_m))^d\right)$ features are required to obtain an $\varepsilon_m$-accurate approximation of the SE kernel. In contrast, Sriperumbudur and Szabó [36] show that for any compact $\mathcal{X} \subset \mathbb{R}^d$, the uniform approximation error using RFF is $\varepsilon_m = O_p(\sqrt{d \ln |\mathcal{X}| / m})$, i.e. at least $O(d \ln |\mathcal{X}| / \varepsilon_m^2)$ features are required to obtain an $\varepsilon_m$-accurate approximation of $k$. In most of the BO applications either $d = O(1)$, or there are enough structure (e.g. generalized additive models) such that effective dimensionality of the problem is low. In that case $O(1/\varepsilon_m^2)$ and $O((\ln(1/\varepsilon_m)^d)$ features are needed to obtain $\varepsilon_m$-accuracy with RFF and QFF approximations, respectively.

Now, recall that the posterior mean and variance of a GP prior $GP_{\mathcal{X}}(0, k)$ with iid Gaussian noise $\mathcal{N}(0, \lambda)$ are given by $\mu_t(x) = k_t(x)^T (K_t + \lambda I_t)^{-1} Y_t$ and $\sigma_t^2(x) = k(x,x) - k_t(x)^T (K_t + \lambda I_t)^{-1} k_t(x)$, respectively. Let $\alpha_t(x) = k_t(x)^T (K_t + \lambda I_t)^{-1} f_t$ denotes the expected posterior mean and $\tilde{\alpha}_t(x) = \tilde{k}_t(x)^T (\tilde{K}_t + \lambda I_t)^{-1} f_t$ denotes the approximation of $\alpha_t(x)$, where $\tilde{k}_t(x) = \tilde{\Phi}_t \tilde{\varphi}(x)$ and $\tilde{K}_t = \tilde{\Phi}_t \tilde{\Phi}_t^T$. Define $\tilde{k}(x,y) = \tilde{\varphi}(x)^T \tilde{\varphi}(y)$. Then, the approximate posterior variance under QFF approximation is $\tilde{\sigma}_t^2(x) = \lambda \tilde{\varphi}_t(x)^T \tilde{V}_t^{-1} \tilde{\varphi}_t(x) = \tilde{k}(x,x) - \tilde{k}_t(x)^T (\tilde{K}_t + \lambda I_t)^{-1} \tilde{k}_t(x)$. Now, we will show that the error introduced by uniform approximation reflects in the approximation of the posterior variance and the expected posterior mean.

**Lemma 15 (Error in posterior mean and variance approximations)** *Let $f \in \mathcal{H}_k(\mathcal{X})$, $\|f\|_{\mathcal{H}} \leq B$ and $k(x,x) \leq 1$ for all $x \in \mathcal{X}$. Let $\tilde{\varphi} : \mathcal{X} \to \mathbb{R}^m$ be a feature map such that 23 holds for some $\varepsilon_m < 1$, and $\tilde{\varphi}(x)^T \tilde{\varphi}(y) \leq 1$ for all $x, y \in \mathcal{X}$. Then for all $x \in \mathcal{X}$ and $t \geq 1$, we have*

$$(i) \quad |\alpha_t(x) - \tilde{\alpha}_t(x)| = O(B\varepsilon_m t^2/\lambda) \quad and \quad (ii) \quad |\sigma_t(x) - \tilde{\sigma}_t(x)| = O(\varepsilon_m^{1/2} t/\lambda).$$

**Proof** This proof is inspired from [26], with some notable changes. First, observe that

$$\left| k_t(x)^T (K_t + \lambda I_t)^{-1} f_t - \tilde{k}_t(x)^T (\tilde{K}_t + \lambda I_t)^{-1} f_t \right|$$

$$\overset{(a)}{\leq} \left| \left( k_t(x) - \tilde{k}_t(x) \right)^T (K_t + \lambda I_t)^{-1} f_t \right| + \left| \tilde{k}_t(x)^T \left( (K_t + \lambda I_t)^{-1} - (\tilde{K}_t + \lambda I_t)^{-1} \right) f_t \right|$$

$$\overset{(b)}{\leq} \left\| k_t(x) - \tilde{k}_t(x) \right\|_2 \left\| (K_t + \lambda I_t)^{-1} f_t \right\|_2 + \left\| \tilde{k}_t(x) \right\|_2 \left\| \left( (K_t + \lambda I_t)^{-1} - (\tilde{K}_t + \lambda I_t)^{-1} \right) f_t \right\|_2$$

$$\overset{(c)}{\leq} \left\| k_t(x) - \tilde{k}_t(x) \right\|_2 \left\| (K_t + \lambda I_t)^{-1} \right\|_2 \|f_t\|_2 + \left\| \tilde{k}_t(x) \right\|_2 \left\| (K_t + \lambda I_t)^{-1} - (\tilde{K}_t + \lambda I_t)^{-1} \right\|_2 \|f_t\|_2,$$

where $(a)$ uses triangle inequality, $(b)$ uses Cauchy-Schwartz inequality and $(c)$ uses the definition of operator norm. By our hypothesis, $\|f_t\|_2 \leq Bt^{1/2}$, $\left\| \tilde{k}_t(x) \right\|_2 \leq t^{1/2}$ and $\left\| k_t(x) - \tilde{k}_t(x) \right\|_2 \leq \varepsilon_m t^{1/2}$. Now

$$\left\| (K_t + \lambda I_t)^{-1} - (\tilde{K}_t + \lambda I_t)^{-1} \right\|_2 = \left\| (K_t + \lambda I_t)^{-1} \left( (\tilde{K}_t + \lambda I_t) - (K_t + \lambda I_t) \right) (\tilde{K}_t + \lambda I_t)^{-1} \right\|_2$$

$$= \left\| (K_t + \lambda I_t)^{-1} (\tilde{K}_t - K_t)(\tilde{K}_t + \lambda I_t)^{-1} \right\|_2$$

$$\overset{(a)}{\leq} \left\| (K_t + \lambda I_t)^{-1} \right\|_2 \left\| \tilde{K}_t - K_t \right\|_2 \left\| (\tilde{K}_t + \lambda I_t)^{-1} \right\|_2$$

$$\overset{(b)}{\leq} \varepsilon_m t/\lambda^2,$$

where $(a)$ follows from the sub-multiplicative property of operator norm and $(b)$ follows from the facts that $\left\| K_t - \tilde{K}_t \right\|_2 \leq \sqrt{\sum_{1 \leq i,j \leq t} (k(x_i, x_j) - \tilde{k}(x_i, x_j))^2} \leq \varepsilon_m t$, and that for any p.s.d. matrix $A \in \mathbb{R}^{t \times t}$, $\left\| (A + \lambda I_t)^{-1} \right\|_2 = \lambda_{\max}\{(A + \lambda I_t)^{-1}\} = 1/\lambda_{\min}\{A + \lambda I_t\} \leq 1/\lambda$. Therefore, for all $x \in \mathcal{X}$ and $t \geq 1$, we have

$$|\alpha_t(x) - \tilde{\alpha}_t(x)| \leq \left( \varepsilon_m t^{1/2}/\lambda + \varepsilon_m t^{3/2}/\lambda^2 \right) Bt^{1/2} = O(B\varepsilon_m t^2/\lambda).$$

Now, since $\left| k(x,y) - \tilde{k}(x,y) \right| \leq \varepsilon_m$ for all $x, y \in \mathcal{X}$, we have $\tilde{k}_t(x) = k_t(x) + a_t(x)$ where $\|a_t(x)\|_\infty \leq \varepsilon_m$. This implies

$$\left| \sigma_t^2(x) - \tilde{\sigma}_t^2(x) \right|$$

$$= \left| k(x,y) - \tilde{k}(x,y) \right| + \left| \tilde{k}_t(x)^T (\tilde{K}_t + \lambda I_t)^{-1} \tilde{k}_t(x) - k_t(x)^T (K_t + \lambda I_t)^{-1} k_t(x) \right|$$

$$\leq \varepsilon_m + \left| k_t(x)^T \left( (\tilde{K}_t + \lambda I_t)^{-1} - (K_t + \lambda I_t)^{-1} \right) k_t(x) \right| + 2 \left| a_t(x)^T (\tilde{K}_t + \lambda I_t)^{-1} k_t(x) \right|$$

$$+ \left| a_t(x)^T (\tilde{K}_t + \lambda I_t)^{-1} a_t(x) \right|$$

$$\overset{(a)}{\leq} \varepsilon_m + \left\| (\tilde{K}_t + \lambda I_t)^{-1} - (K_t + \lambda I_t)^{-1} \right\|_2 \|k_t(x)\|_2^2 + 2 \|a_t(x)\|_2 \left\| (\tilde{K}_t + \lambda I_t)^{-1} \right\|_2 \|k_t(x)\|_2$$

$$+ \left\| (\tilde{K}_t + \lambda I_t)^{-1} \right\|_2 \|a_t(x)\|_2^2$$

$$\overset{(b)}{\leq} \varepsilon_m + \varepsilon_m t^2/\lambda^2 + 2\varepsilon_m t/\lambda + \varepsilon_m^2 t/\lambda = O(\varepsilon_m t^2/\lambda^2) \text{ for } \varepsilon_m < 1.$$

Here $(a)$ is due to Cauchy-Schwartz inequality and definition of operator norm. $(b)$ uses $\|k_t(x)\|_2 \leq t^{1/2}$, $\|a_t(x)\|_2 \leq \varepsilon_m t^{1/2}$, $\left\| (\tilde{K}_t + \lambda I_t)^{-1} - (K_t + \lambda I_t)^{-1} \right\|_2 \leq \varepsilon_m t/\lambda^2$ and $\left\| (\tilde{K}_t + \lambda I_t)^{-1} \right\|_2 \leq 1/\lambda$. Now, the result follows from the fact that for any $a, b \geq 0$, $(a+b)^{1/2} \leq a^{1/2} + b^{1/2}$. ∎

Now, we are ready to prove Lemma 1.

### D.2.2 Proof of Lemma 1

Under the QFF approximation, we have $\tilde{\varphi}_t = \tilde{\varphi}$ and $m_t = m$ for all $t \geq 1$. Hence, we have $\tilde{\mu}_t(x) = \tilde{\varphi}(x)^T \tilde{\theta}_t$ and $\tilde{\alpha}_t(x) = \tilde{\varphi}(x)^T \tilde{\Phi}_t^T (\tilde{\Phi}_t \tilde{\Phi}_t^T + \lambda I_t)^{-1} f_t = \tilde{\varphi}(x)^T \tilde{V}_t^{-1} \tilde{\Phi}_t^T f_t$, where the last equality follows from 3. Now, by Cauchy-Schwartz inequality,

$$|\tilde{\alpha}_t(x) - \tilde{\mu}_t(x)| \leq \left\| \tilde{V}_t^{-1} \tilde{\Phi}^T f_t - \tilde{\theta}_t \right\|_{\tilde{V}_t} \|\tilde{\varphi}(x)\|_{\tilde{V}_t^{-1}} = \lambda^{-1/2} \left\| \tilde{V}_t^{-1} \tilde{\Phi}_t^T f_t - \tilde{\theta}_t \right\|_{\tilde{V}_t} \tilde{\sigma}_t(x).$$

Hence, from Lemma 13, we have, for any $\delta \in (0,1]$, with probability at least $1 - \delta$, uniformly over all $x \in \mathcal{X}$ and $t \in [T]$, that

$$|\tilde{\alpha}_t(x) - \tilde{\mu}_t(x)| \leq 4\sqrt{m/\lambda}\, v^{\frac{1}{1+\alpha}} \left(\ln(2mT/\delta)\right)^{\frac{\alpha}{1+\alpha}} t^{\frac{1-\alpha}{2(1+\alpha)}} \tilde{\sigma}_t(x). \tag{24}$$

By triangle inequality,

$$|f(x) - \tilde{\mu}_t(x)| \leq |f(x) - \alpha_t(x)| + |\alpha_t(x) - \tilde{\alpha}_t(x)| + |\tilde{\alpha}_t(x) - \tilde{\mu}_t(x)|.$$

Now, from 7, $|f(x) - \alpha_t(x)| \leq B\sigma_t(x)$ and thus, in turn, from Lemma 15, $|f(x) - \alpha_t(x)| = B\tilde{\sigma}_t(x) + O(B\varepsilon_m^{1/2} t/\lambda)$. Also, from Lemma 15, $|\alpha_t(x) - \tilde{\alpha}_t(x)| = O(B\varepsilon_m t^2/\lambda)$. Now combining these with 24, we obtain, for any $\delta \in (0,1]$, with probability at least $1 - \delta$, uniformly over all $x \in \mathcal{X}$ and $t \in [T]$, that

$$
\begin{aligned}
|f(x) - \tilde{\mu}_t(x)| &\leq \left(B + 4\sqrt{m/\lambda}\, v^{\frac{1}{1+\alpha}} \left(\ln(2mT/\delta)\right)^{\frac{\alpha}{1+\alpha}} t^{\frac{1-\alpha}{2(1+\alpha)}}\right) \tilde{\sigma}_t(x) + O(B\varepsilon_m^{1/2} t/\lambda) + O(B\varepsilon_m t^2/\lambda) \\
&= \left(B + 4\sqrt{m/\lambda}\, v^{\frac{1}{1+\alpha}} \left(\ln(2mT/\delta)\right)^{\frac{\alpha}{1+\alpha}} t^{\frac{1-\alpha}{2(1+\alpha)}}\right) \tilde{\sigma}_t(x) + O(B\varepsilon_m^{1/2} t^2/\lambda)
\end{aligned}
$$

for $\varepsilon_m < 1$. Further observe that $|f(x) - \tilde{\mu}_0(x)| = |f(x)| \leq Bk^{1/2}(x,x) = B\sigma_0(x) \leq B\tilde{\sigma}_0(x) + B\varepsilon_m^{1/2}$. Now the result follows by setting $\beta_{t+1} = B + 4\sqrt{m/\lambda}\, v^{\frac{1}{1+\alpha}} \left(\ln(2mT/\delta)\right)^{\frac{\alpha}{1+\alpha}} t^{\frac{1-\alpha}{2(1+\alpha)}}$ for all $t \geq 0$.

### D.2.3 Proof of Theorem 3

For any $\delta \in (0,1]$, we have, with probability at least $1 - \delta$, uniformly over all $t \in [T]$, the instantaneous regret

$$
\begin{aligned}
r_t &= f(x^\star) - f(x_t) \\
&\overset{(a)}{\leq} \tilde{\mu}_{t-1}(x^\star) + \beta_t \tilde{\sigma}_{t-1}(x^\star) + O(B\varepsilon_m^{1/2} t^2/\lambda) - f(x_t) \\
&\overset{(b)}{\leq} \tilde{\mu}_{t-1}(x_t) + \beta_t \tilde{\sigma}_{t-1}(x_t) - f(x_t) + O(B\varepsilon_m^{1/2} t^2/\lambda) \\
&\overset{(c)}{\leq} 2\beta_t \tilde{\sigma}_{t-1}(x_t) + O(B\varepsilon_m^{1/2} t^2/\lambda).
\end{aligned}
$$

Here $(a)$ and $(c)$ follow from Lemma 1 and $(b)$ is due to the choice of ATA-GP-UCB (Algorithm 2). Now Observe that $(\beta_t)_{t \geq 1}$ is an increasing sequence in $t$. Further,

$$\sum_{t=1}^{T} \tilde{\sigma}_{t-1}(x_t) \overset{(a)}{\leq} \sqrt{T \sum_{t=1}^{T} \tilde{\sigma}_{t-1}^2(x_t)} \overset{(b)}{\leq} \sqrt{2(1+\lambda)T\tilde{\gamma}_T} = O(\sqrt{mT \ln T}).$$

Here $(a)$ follows from Cauchy-Schwartz inequality, $(b)$ from Lemma 6, and $(c)$ from Lemma 5 noting that $\tilde{k}$ is a linear kernel defined on $\mathbb{R}^{2m}$. Hence for any $\delta \in (0,1]$, with probability at least $1 - \delta$, the cumulative regret of ATA-GP-UCB after $T$ rounds is

$$
\begin{aligned}
R_T &= O\left(\beta_T \sqrt{Tm \ln T}\right) + \sum_{t=1}^{T} O(B\varepsilon_m^{1/2} t^2/\lambda) \\
&= O\left(B\sqrt{Tm \ln T} + mv^{\frac{1}{1+\alpha}} \left(\ln(mT/\delta)\right)^{\frac{\alpha}{1+\alpha}} (\ln T)^{1/2} T^{\frac{1}{1+\alpha}} + B\varepsilon_m^{1/2} T^3\right).
\end{aligned}
$$

From Lemma 14 if $\bar{m} \geq 1/l^2$ and $d = O(1)$, we have $\varepsilon_m = O((e/4)^{\bar{m}})$. Further if $\bar{m} \geq \log_{4/e}(T^6)$, then $\varepsilon_m^{1/2} T^3 = O(1)$. Now choosing $\bar{m} = \Theta\left(\log_{4/e}(T^6)\right)$, we can ensure that $m = O((\ln T)^d)$ [6].

Therefore for any $\delta \in (0, 1]$, with probability at least $1 - \delta$, the cumulative regret of ATA-GP-UCB under QFF approximation after $T$ rounds is

$$R_T = O\left(B\sqrt{T(\ln T)^{d+1}} + v^{\frac{1}{1+\alpha}}\left(\ln\left(T(\ln T)^d/\delta\right)\right)^{\frac{\alpha}{1+\alpha}}\sqrt{\ln T}(\ln T)^d T^{\frac{1}{1+\alpha}}\right).$$

### D.3 Analysis of ATA-GP-UCB under Nyström approximation

#### D.3.1 Construction of dictionary and its properties

Given the kernel matrix $K_t$, we define an accurate dictionary as follows.

**Definition 3 ($\varepsilon$-accurate dictionary [9])** *For any $\varepsilon \in (0, 1)$, a dictionary $\mathcal{D}_t \subseteq \{x_1, \ldots, x_t\}$ is said to be $\varepsilon$-accurate with respect to the kernel matrix $K_t$ if*

$$\left\|(K_t + \lambda I)^{-1/2}K_t^{1/2}(I_t - S_t^2)K_t^{1/2}(K_t + \lambda I)^{-1/2}\right\|_2 \le \varepsilon,$$

*where $S_t$ is the selection matrix associated with the dictionary $\mathcal{D}_t$ such that $[S_t]_{i,i} = 1/\sqrt{p_{t,i}}$ if $x_i \in \mathcal{D}_t$, and $0$, elsewhere.*

The following lemma states two more equivalent condition for a dictionary to be accurate.

**Lemma 16** *Let $V_{\mathcal{D}_t} = \Phi_t^T S_t^2 \Phi_t + \lambda I_{\mathcal{H}}$. Then, the following are equivalent:*

1. $\left\|(K_t + \lambda I)^{-1/2}K_t^{1/2}(I_t - S_t^2)K_t^{1/2}(K_t + \lambda I)^{-1/2}\right\|_2 \le \varepsilon$.

2. $\left\|(\Phi_t^T \Phi_t + \lambda I_{\mathcal{H}})^{-1/2}\Phi_t^T(I_t - S_t^2)\Phi_t(\Phi_t^T \Phi_t + \lambda I_{\mathcal{H}})^{-1/2}\right\|_{\mathcal{H}} \le \varepsilon$.

3. $(1 - \varepsilon)V_t \preceq V_{\mathcal{D}_t} \preceq (1 + \varepsilon)V_t$.

**Proof** Let $\Phi_t = U\Sigma V^T$ be the singular value decomposition of $\Phi_t$. Then $\Phi_t(\Phi_t^T \Phi_t + \lambda I_{\mathcal{H}})^{-1/2} = U\Sigma(\Sigma^T \Sigma + \lambda I_{\mathcal{H}})^{-1}V^T$, $(\Phi_t^T \Phi_t + \lambda I_{\mathcal{H}})^{-1/2}\Phi_t^T = V(\Sigma^T \Sigma + \lambda I_{\mathcal{H}})^{-1}\Sigma^T U^T$ and $K_t = U\Sigma\Sigma^T U^T$. Therefore

$$\left\|(\Phi_t^T \Phi_t + \lambda I_{\mathcal{H}})^{-1/2}\Phi_t^T(I_t - S_t^2)\Phi_t(\Phi_t^T \Phi_t + \lambda I_{\mathcal{H}})^{-1/2}\right\|_{\mathcal{H}}$$

$$= \left\|V(\Sigma^T \Sigma + \lambda I_{\mathcal{H}})^{-1/2}\Sigma^T U^T(I_t - S_t^2)U\Sigma(\Sigma^T \Sigma + \lambda I_{\mathcal{H}})^{-1/2}V^T\right\|_{\mathcal{H}}$$

$$= \left\|(\Sigma^T \Sigma + \lambda I_{\mathcal{H}})^{-1/2}\Sigma^T U^T(I_t - S_t^2)U\Sigma(\Sigma^T \Sigma + \lambda I_{\mathcal{H}})^{-1/2}\right\|_{\mathcal{H}}$$

$$= \left\|(\Sigma\Sigma^T + \lambda I_t)^{-1/2}(\Sigma\Sigma^T)^{1/2}U^T(I_t - S_t^2)U(\Sigma\Sigma^T)^{1/2}(\Sigma\Sigma^T + \lambda I_t)^{-1/2}\right\|_2$$

$$= \left\|U(\Sigma\Sigma^T + \lambda I_t)^{-1/2}(\Sigma\Sigma^T)^{1/2}U^T(I_t - S_t^2)U(\Sigma\Sigma^T)^{1/2}(\Sigma\Sigma^T + \lambda I_t)^{-1/2}U^T\right\|_2$$

$$= \left\|(K_t + \lambda I)^{-1/2}K_t^{1/2}(I_t - S_t^2)K_t^{1/2}(K_t + \lambda I)^{-1/2}\right\|_2,$$

which proves that $1 \iff 2$. Now, Observe that

$$\left\|(\Phi_t^T \Phi_t + \lambda I_{\mathcal{H}})^{-1/2}\Phi_t^T(I_t - S_t^2)\Phi_t(\Phi_t^T \Phi_t + \lambda I_{\mathcal{H}})^{-1/2}\right\|_{\mathcal{H}} \le \varepsilon$$

$$\iff -\varepsilon I_{\mathcal{H}} \preceq (\Phi_t^T \Phi_t + \lambda I_{\mathcal{H}})^{-1/2}(\Phi_t^T \Phi_t - \Phi_t^T S_t^2 \Phi_t)(\Phi_t^T \Phi_t + \lambda I_{\mathcal{H}})^{-1/2} \preceq \varepsilon I_{\mathcal{H}}$$

$$\iff -\varepsilon I_{\mathcal{H}} \preceq V_t^{-1/2}(V_t - V_{\mathcal{D}_t})V_t^{-1/2} \preceq \varepsilon I_{\mathcal{H}}$$

$$\iff -\varepsilon V_t \preceq V_t - V_{\mathcal{D}_t} \preceq \varepsilon V_t$$

$$\iff (1 - \varepsilon)V_t \preceq V_{\mathcal{D}_t} \preceq (1 + \varepsilon)V_t,$$

which proves $2 \iff 3$. ∎

An $\varepsilon$-accurate dictionary can be obtained by including points proportional to their $\lambda$-ridge leverage scores defined as follows.

**Definition 4 (Ridge leverage score [1])** *For a set of points $\{x_1, \ldots, x_t\}$ and a constant $\lambda > 0$, the $\lambda$- ridge leverage score of the point $x_i, i \in [t]$ is defined as*

$$l_{t,i} = e_i^T K_t (K_t + \lambda I_t)^{-1} e_i,$$

*where $e_i \in \mathbb{R}^t$ is the $i$-th standard basis vector.*

Ridge leverage score (RLS) can be interpreted in many ways and it is well studied in the literature. Here we observe that

$$e_i^T K_t (K_t + \lambda I_t)^{-1} e_i = e_i^T \Phi_t \Phi_t^T (\Phi_t \Phi_t^T + \lambda I_t)^{-1} e_i = e_i^T \Phi_t (\Phi_t^T \Phi_t + \lambda I_{\mathcal{H}})^{-1} \Phi_t^T e_i = \|\varphi(x_i)\|_{V_t^{-1}}^2.$$

Therefore $l_{t,i} = \frac{1}{\lambda} \sigma_t^2(x_i)$, i.e., the RLS of $x_i$ is proportional its posterior variance $\sigma_t^2(x_i)$ under the GP prior $GP_{\mathcal{X}}(0, k)$. However, the exact computation of $\lambda$-ridge leverage scores in turn requires inverting the kernel matrix $K_t$ which requires $O(t^3)$ time. This motivates the need for a fast approximation of RLS such that it can be used to construct an $\varepsilon$-accurate dictionary. Calandriello et al. [9] show that, instead of using the exact ridge leverage scores (or, equivalently, posterior variances) if we use the approximate variances from the previous round to sample points in the current round, then we will be able to obtain an accurate dictionary. Not only that, the dictionary size will grow no faster than the maximum information gain of the underlying kernel. Now, we present the NyströmEmbedding procedure which is used in Algorithm 2.

---

**Algorithm 3** NyströmEmbedding

> **Input:** $\{(x_i, \tilde{\sigma}_{t-1}(x_i))\}_{i=1}^t, q$
> **Set:** $\mathcal{D}_t = \emptyset$
> **for** $i = 1, 2, 3 \ldots, t$ **do**
>      Sample $z_{t,i} \sim \mathcal{B}\left(\min\{q\tilde{\sigma}_{t-1}^2(x_i), 1\}\right)$
>      If $z_{t,i} = 1$, set $\mathcal{D}_t = \mathcal{D}_t \cup \{x_i\}$
> **end for**
> **Return** $\tilde{\varphi}_t(x) = \left(K_{\mathcal{D}_t}^{1/2}\right)^+ k_{\mathcal{D}_t}(x)$

---

The following lemma states the properties of the dictionaries $\mathcal{D}_t$ constructed using Algorithm 3.

**Lemma 17 (Properties of the dictionary)** *For any $\varepsilon \in (0, 1)$ and $\delta \in (0, 1]$, set $\rho = \frac{1+\varepsilon}{1-\varepsilon}$ and $q = \frac{6\rho \ln(2T/\delta)}{\varepsilon^2}$. Then, with probability at least $1 - \delta$, uniformly over all $t \in [T]$,*

$$(1 - \varepsilon)V_t \preceq V_{\mathcal{D}_t} \preceq (1 + \varepsilon)V_t \quad \text{and} \quad m_t \leq 6\rho\left(1 + \frac{1}{\lambda}\right) q\gamma_t.$$

Lemma 17 is a restatement of [9, Theorem 1] and it is presented in this form for the sake of brevity and completeness. Now, we will show that using the Nyström embeddings $\tilde{\varphi}_t(x)$, we can prevent the variance starvation which generally arises due to approximation.

### D.3.2    Preventing variance starvation with Nyström embeddings

Recall that the posterior mean and variance of a GP prior $GP_{\mathcal{X}}(0, k)$ with iid Gaussian noise $\mathcal{N}(0, \lambda)$ are given by $\mu_t(x) = k_t(x)^T (K_t + \lambda I_t)^{-1} Y_t$ and $\sigma_t^2(x) = k(x, x) - k_t(x)^T (K_t + \lambda I_t)^{-1} k_t(x)$, respectively. Let $\alpha_t(x) = k_t(x)^T (K_t + \lambda I_t)^{-1} f_t$ denotes the expected posterior mean and $\tilde{\alpha}_t(x) = \tilde{k}_t(x)^T (\tilde{K}_t + \lambda I_t)^{-1} f_t$ denotes the approximation of $\alpha_t(x)$, where $\tilde{k}_t(x) = \tilde{\Phi}_t \tilde{\varphi}(x)$ and $\tilde{K}_t = \tilde{\Phi}_t \tilde{\Phi}_t^T$. Then, we have $\alpha_t(x) = \langle \varphi(x), V_t^{-1} \Phi_t^T f_t \rangle_{\mathcal{H}}$ and $\tilde{\alpha}_t(x) = \tilde{\varphi}_t(x)^T \tilde{V}_t^{-1} \tilde{\Phi}_t^T f_t$. Now, we can rewrite the posterior variance as $\sigma_t^2(x) = \lambda \|\varphi(x)\|_{V_t^{-1}}^2$, whereas the approximate posterior variance under Nyström approximation is given by $\tilde{\sigma}_t^2(x) = k(x, x) - \tilde{\varphi}_t(x)^T \tilde{\varphi}_t(x) + \lambda \tilde{\varphi}_t(x)^T \tilde{V}_t^{-1} \tilde{\varphi}_t(x)$. This choice of $\tilde{\sigma}_t^2(x)$ helps us to negate the variance starvation which arises due to feature approximation. Now, we will justify this choice of $\tilde{\sigma}_t^2(x)$ by showing that it can be derived by projecting $\varphi(x)$ to a smaller RKHS. The idea is inspired from Calandriello et al. [9].

**Projection to a smaller RKHS:** For any dictionary $\mathcal{D}_t = \{x_{i_1}, \ldots, x_{i_{m_t}}\}, i_j \in [t]$, define the operator $\Phi_{\mathcal{D}_t} : \mathcal{H}_k(\mathcal{X}) \to \mathbb{R}^{m_t}$ such that for any $h \in \mathcal{H}_k(\mathcal{X})$, $\Phi_{\mathcal{D}_t} h =$

$\left[\langle\varphi(x_{i_1}),h\rangle_{\mathcal{H}},\ldots,\langle\varphi(x_{i_{m_t}}),h\rangle_{\mathcal{H}}\right]^T$ and denote its adjoint by $\Phi_{\mathcal{D}_t}^T : \mathbb{R}^{m_t} \to \mathcal{H}_k(\mathcal{X})$. Let $\widehat{\varphi}_t(x) = P_t\varphi(x)$ be the projection of $\varphi(x)$ to the subspace spanned by the columns of the operator $\Phi_{\mathcal{D}_t}^T$, where the projection operator $P_t : \mathcal{H}_k(\mathcal{X}) \to \mathrm{Col}(\Phi_{\mathcal{D}_t}^T)$ is given by $P_t = \Phi_{\mathcal{D}_t}^T(\Phi_{\mathcal{D}_t}\Phi_{\mathcal{D}_t}^T)^+\Phi_{\mathcal{D}_t}$. It is easy to see that $P_t^T = P_t$ and $P_t^2 = P_t$. Now, for any set $\{x_1,\ldots,x_t\} \subset \mathcal{X}$ define the operator $\widehat{\Phi}_t : \mathcal{H}_k(\mathcal{X}) \to \mathbb{R}^t$ such that for any $h \in \mathcal{H}_k(\mathcal{X})$, $\widehat{\Phi}_t h = [\langle\widehat{\varphi}_t(x_1),h\rangle_{\mathcal{H}},\ldots,\langle\widehat{\varphi}_t(x_t),h\rangle_{\mathcal{H}}]^T$, and denote its adjoint by $\widehat{\Phi}_t^T : \mathbb{R}^t \to \mathcal{H}_k(\mathcal{X})$.

**Lemma 18 (Approximate posterior variance and mean under projection)** *Let* $\widehat{V}_t = \widehat{\Phi}_t^T\widehat{\Phi}_t + \lambda I_{\mathcal{H}}$ *for any* $\lambda > 0$*. Then, we have*

$$\tilde{\sigma}_t^2(x) = \lambda\,\|\varphi(x)\|_{\widehat{V}_t^{-1}}^2 \quad and \quad \tilde{\alpha}_t(x) = \langle\varphi(x),\widehat{V}_t^{-1}\widehat{\Phi}_t^T f_t\rangle_{\mathcal{H}}.$$

**Proof** Since $K_{\mathcal{D}_t} = \Phi_{\mathcal{D}_t}\Phi_{\mathcal{D}_t}^T$, we have the projection $P_t = \Phi_{\mathcal{D}_t}^T(K_{\mathcal{D}_t})^+\Phi_{\mathcal{D}_t}$. Now, observe that $\langle\varphi(x),\varphi(y)\rangle_{P_t} = \left((K_{\mathcal{D}_t}^{1/2})^\dagger\Phi_{\mathcal{D}_t}\varphi(x)\right)^T\left((K_{\mathcal{D}_t}^{1/2})^\dagger\Phi_{\mathcal{D}_t}\varphi(y)\right) = \left((K_{\mathcal{D}_t}^{1/2})^\dagger k_{\mathcal{D}_t}(x)\right)^T\left((K_{\mathcal{D}_t}^{1/2})^\dagger k_{\mathcal{D}_t}(y)\right) = \tilde{\varphi}_t(x)^T\tilde{\varphi}_t(y)$. Also, note that $\widehat{\Phi}_t^T = P_t\Phi_t^T$. This implies $\widehat{\Phi}_t\varphi(x) = \Phi_tP_t\varphi(x) = [\langle\varphi(x_1),\varphi(x)\rangle_{P_t},\ldots,\langle\varphi(x_t),\varphi(x)\rangle_{P_t}]^T = [\tilde{\varphi}_t(x_1)^T\tilde{\varphi}_t(x),\ldots,\tilde{\varphi}_t(x_t)^T\tilde{\varphi}_t(x)]^T = \tilde{\Phi}_t\tilde{\varphi}_t(x)$. Further, the $(i,j)$-th entry of $\widehat{\Phi}_t\widehat{\Phi}_t^T$ is given by $[\widehat{\Phi}_t\widehat{\Phi}_t^T]_{i,j} = \langle P_t\varphi(x_i),P_t\varphi(x_j)\rangle_{\mathcal{H}} = \langle\varphi(x_i),\varphi(x_j)\rangle_{P_t} = \tilde{\varphi}_t(x_i)^T\tilde{\varphi}_t(x_j)$ and hence, $\widehat{\Phi}_t\widehat{\Phi}_t^T = \tilde{\Phi}_t\tilde{\Phi}_t^T$. Then, we have

$$
\begin{aligned}
\lambda\,\|\varphi(x)\|_{\widehat{V}_t^{-1}}^2 &= \lambda\langle\varphi(x),(\widehat{\Phi}_t^T\widehat{\Phi}_t + \lambda I_{\mathcal{H}})^{-1}\varphi(x)\rangle_{\mathcal{H}} \\
&\overset{(a)}{=} \langle\varphi(x),\left(I_{\mathcal{H}} - \widehat{\Phi}_t^T(\widehat{\Phi}_t\widehat{\Phi}_t^T + \lambda I_t)^{-1}\widehat{\Phi}_t\right)\varphi(x)\rangle_{\mathcal{H}} \\
&\overset{(b)}{=} k(x,x) - \tilde{\varphi}_t(x)^T\tilde{\Phi}_t^T(\tilde{\Phi}_t\tilde{\Phi}_t^T + \lambda I_t)^{-1}\tilde{\Phi}_t\tilde{\varphi}_t(x) \\
&\overset{(c)}{=} k(x,x) - \tilde{\varphi}_t(x)^T\tilde{\Phi}_t^T\tilde{\Phi}_t(\tilde{\Phi}_t^T\tilde{\Phi}_t + \lambda I_{m_t})^{-1}\tilde{\varphi}_t(x) \\
&= k(x,x) - \tilde{\varphi}_t(x)^T\tilde{\varphi}_t(x) + \lambda\tilde{\varphi}_t(x)^T\tilde{V}_t^{-1}\tilde{\varphi}_t(x) = \tilde{\sigma}_t^2(x).
\end{aligned}
$$

Here $(a)$ follows from 4, $(b)$ is due to $\widehat{\Phi}_t\varphi(x) = \tilde{\Phi}_t\tilde{\varphi}_t(x)$ and $\widehat{\Phi}_t\widehat{\Phi}_t^T = \tilde{\Phi}_t\tilde{\Phi}_t^T$, and $(c)$ follows from 3. Now observe that

$$
\begin{aligned}
\langle\varphi(x),\widehat{V}_t^{-1}\widehat{\Phi}_t^T f_t\rangle_{\mathcal{H}} &= \langle\varphi(x),(\widehat{\Phi}_t^T\widehat{\Phi}_t + \lambda I_{\mathcal{H}})^{-1}\widehat{\Phi}_t^T f_t\rangle_{\mathcal{H}} \\
&\overset{(a)}{=} \langle\varphi(x),\widehat{\Phi}_t^T(\widehat{\Phi}_t\widehat{\Phi}_t^T + \lambda I_t)^{-1}f_t\rangle_{\mathcal{H}} \\
&\overset{(b)}{=} \tilde{\varphi}_t(x)^T\tilde{\Phi}_t^T(\tilde{\Phi}_t\tilde{\Phi}_t^T + \lambda I_t)^{-1}f_t \\
&\overset{(c)}{=} \tilde{\varphi}_t(x)^T(\tilde{\Phi}_t^T\tilde{\Phi}_t + \lambda I_{m_t})^{-1}\tilde{\Phi}_t^T f_t \\
&= \tilde{\varphi}_t(x)^T\tilde{V}_t^{-1}\tilde{\Phi}_t^T f_t = \tilde{\alpha}_t(x).
\end{aligned}
$$

Here $(a)$ and $(c)$ follow from 3, and $(b)$ is due to $\widehat{\Phi}_t\varphi(x) = \tilde{\Phi}_t\tilde{\varphi}_t(x)$ and $\widehat{\Phi}_t\widehat{\Phi}_t^T = \tilde{\Phi}_t\tilde{\Phi}_t^T$. ∎

**Lemma 19 (Accuracy of approximate posterior variance)** *For an $\varepsilon$-accurate dictionary (Definition 3), we have*

$$\frac{1-\varepsilon}{1+\varepsilon}\sigma_t^2(x) \le \tilde{\sigma}_t^2(x) \le \frac{1+\varepsilon}{1-\varepsilon}\sigma_t^2(x).$$

**Proof** From Lemma 18, we have $\tilde{\sigma}_t^2(x) = \lambda\langle\varphi(x),\widehat{V}_t^{-1}\varphi(x)\rangle_{\mathcal{H}}$. Now, observe that $\widehat{V}_t = P_t\Phi_t^T\Phi_tP_t + \lambda I_{\mathcal{H}} = P_tV_tP_t + \lambda(I_{\mathcal{H}} - P_t)$. From Lemma 16, we have $(1-\varepsilon)V_t \preceq V_{\mathcal{D}_t} \preceq (1+\varepsilon)V_t$

for any $\varepsilon$-accurate dictionary $\mathcal{D}_t$. This implies that

$$
\begin{aligned}
\widehat{V}_t &\preceq \frac{1}{1-\varepsilon} P_t V_{\mathcal{D}_t} P_t + \lambda(I_\mathcal{H} - P_t) \\
&= \frac{1}{1-\varepsilon} P_t \Phi_t^T S_t^2 \Phi_t P_t + \frac{\lambda\varepsilon}{1-\varepsilon} P_t + \lambda I_\mathcal{H} \\
&\overset{(a)}{\preceq} \frac{1}{1-\varepsilon} (\Phi_t^T S_t^2 \Phi_t + \lambda I_\mathcal{H}) \\
&\preceq \frac{1+\varepsilon}{1-\varepsilon} V_t,
\end{aligned}
$$

where $(a)$ follows from $P_t \Phi_t^T S_t = \Phi_t^T S_t$ and $P_t \preceq I_\mathcal{H}$. Therefore, we have

$$
\tilde{\sigma}_t^2(x) \geq \frac{1-\varepsilon}{1+\varepsilon} \lambda \langle \varphi(x), V_t^{-1} \varphi(x) \rangle_\mathcal{H} = \frac{1-\varepsilon}{1+\varepsilon} \sigma_t^2(x).
$$

Similarly, we can show that $\widehat{V}_t \succeq \frac{1-\varepsilon}{1+\varepsilon} V_t$ and thus, in turn, $\tilde{\sigma}_t^2(x) \leq \frac{1+\varepsilon}{1-\varepsilon} \sigma_t^2(x)$. ∎

Now, we will show that the confidence sets formed by ATA-GP-UCB (Algorithm 2) under Nyström approximation is tighter compared to that of TGP-UCB.

### D.3.3 Confidence sets of ATA-GP-UCB under Nyström approximation

First, we define the following two events. Fix any $\varepsilon \in (0,1)$ and $\delta \in (0,1]$. Let $E_{1,t}$ denotes the event that the dictionary $\mathcal{D}_t$ is $\varepsilon$-accurate, i.e,

$$
(1-\varepsilon)V_t \preceq V_{\mathcal{D}_t} \preceq (1+\varepsilon)V_t,
$$

and $E_{2,t}$ denotes the event that the size of the dictionary $\mathcal{D}_t$ is at most $6\rho(1+\frac{1}{\lambda})q\gamma_t$, i.e.,

$$
m_t \leq 6\rho \left(1 + \frac{1}{\lambda}\right) q\gamma_t,
$$

where $\rho = \frac{1+\varepsilon}{1-\varepsilon}$ and $q = \frac{6\rho \ln(2T/\delta)}{\varepsilon^2}$. Then from Lemma 17, we have $\mathbb{P}\left[\cap_{t=1}^T (E_{1,t} \cap E_{2,t})\right] \geq 1-\delta$. Let $\mathcal{G}_t = \sigma\left(\{x_i, (z_{i,j})_{j=1}^i\}_{i=1}^t\right), t \geq 1$ denotes the $\sigma$-algebra generated by the arms played and the outcomes of the NyströmEmbedding procedure(Algorithm 3) up to time $t$. See that $(\mathcal{G}_t)_{t\geq 1}$ defines a filtration, and both $E_{1,t}$ and $E_{2,t}$ are $\mathcal{G}_t$ measurable.

**Lemma 20 (Tighter confidence sets with Nyström embedding)** *Fix any $\delta \in (0,1]$, $\varepsilon \in (0,1)$ and set $\rho = \frac{1+\varepsilon}{1-\varepsilon}$. Then, ATA-GP-UCB under Nyström approximation, and with parameters $q = 6\rho \ln(4T/\delta)/\varepsilon^2$, $b_t = (v/\ln(4m_tT/\delta))^{\frac{1}{1+\alpha}} t^{\frac{1-\alpha}{2(1+\alpha)}}$ and $\beta_{t+1} = B(1 + \frac{1}{\sqrt{1-\varepsilon}}) + 4\sqrt{m_t/\lambda}\, v^{\frac{1}{1+\alpha}} (\ln(4m_tT/\delta))^{\frac{\alpha}{1+\alpha}} t^{\frac{1-\alpha}{2(1+\alpha)}}$, ensures, with probability at least $1-\delta$, uniformly over all $t \in [T]$ and $x \in \mathcal{X}$, that*

$$
|f(x) - \tilde{\mu}_{t-1}(x)| \leq \beta_t \tilde{\sigma}_{t-1}(x),
$$

*where $m_t$ is the dimension of the Nyström embedding $\tilde{\varphi}_t$ constructed at round $t$.*

**Proof** From Lemma 18, we have $\tilde{\alpha}_t(x) = \langle \varphi(x), \widehat{V}_t^{-1} \widehat{\Phi}_t^T f_t \rangle_\mathcal{H}$. Therefore,

$$
\begin{aligned}
|f(x) - \tilde{\alpha}_t(x)| &= \left|\langle \varphi(x), f - \widehat{V}_t^{-1} \widehat{\Phi}_t^T f_t \rangle_\mathcal{H}\right| \\
&\overset{(a)}{\leq} \|\varphi(x)\|_{\widehat{V}_t^{-1}} \left\|f - \widehat{V}_t^{-1} \widehat{\Phi}_t^T f_t\right\|_{\widehat{V}_t} \\
&= \lambda^{-1/2} \left\|(\widehat{\Phi}_t^T \widehat{\Phi}_t + \lambda I_\mathcal{H})f - \widehat{\Phi}_t^T \Phi_t f\right\|_{\widehat{V}_t^{-1}} \tilde{\sigma}_t(x) \\
&\overset{(b)}{=} \lambda^{-1/2} \left\|\lambda f - \widehat{\Phi}_t^T \Phi_t (I_\mathcal{H} - P_t)f\right\|_{\widehat{V}_t^{-1}} \tilde{\sigma}_t(x) \\
&\overset{(c)}{\leq} \left(\lambda^{1/2} \left\|\widehat{V}_t^{-1/2} f\right\|_\mathcal{H} + \lambda^{-1/2} \left\|\widehat{V}_t^{-1/2} \widehat{\Phi}_t^T \Phi_t (I_\mathcal{H} - P_t)f\right\|_\mathcal{H}\right) \tilde{\sigma}_t(x) \\
&\overset{(d)}{\leq} \left(\|f\|_\mathcal{H} + \lambda^{-1/2} \left\|\widehat{V}_t^{-1/2} \widehat{\Phi}_t^T\right\|_\mathcal{H} \|\Phi_t(I_\mathcal{H} - P_t)\|_\mathcal{H} \|f\|_\mathcal{H}\right) \tilde{\sigma}_t(x) \\
&\overset{(e)}{\leq} B\left(1 + \lambda^{-1/2} \|\Phi_t(I_\mathcal{H} - P_t)\|_\mathcal{H}\right) \tilde{\sigma}_t(x).
\end{aligned}
$$

Here $(a)$ is by Cauchy-Schwartz inequality, $(b)$ uses the fact that $\widehat{\Phi}_t = \Phi_t P_t$, $(c)$ is by triangle inequality, $(d)$ follows from $\left\|\widehat{V}_t^{-1/2} f\right\|_{\mathcal{H}} \leq \lambda^{-1/2} \|f\|_{\mathcal{H}}$, and $(e)$ follows from the fact that $\left\|\widehat{V}_t^{-1/2} \widehat{\Phi}_t^T\right\|_{\mathcal{H}}^2 = \lambda_{\max}\left(\widehat{\Phi}_t(\widehat{\Phi}_t^T \widehat{\Phi}_t + \lambda I_{\mathcal{H}})^{-1}\widehat{\Phi}_t^T\right) = \lambda_{\max}\left(\widehat{\Phi}_t\widehat{\Phi}_t^T(\widehat{\Phi}_t\widehat{\Phi}_t^T + \lambda I_t)^{-1}\right) \leq 1$, and that $\|f\|_{\mathcal{H}} \leq B$. Now see that $\mathrm{Col}(\Phi_{\mathcal{D}_t}^T) = \mathrm{Col}(\Phi_t^T S_t)$, and hence $P_t = \Phi_t^T S_t(S_t \Phi_t \Phi_t^T S_t)^+ S_t \Phi_t$. Therefore

$$I_{\mathcal{H}} - P_t \preceq I_{\mathcal{H}} - \Phi_t^T S_t(S_t\Phi_t\Phi_t^T S_t + \lambda I_{\mathcal{H}})^{-1}S_t\Phi_t \overset{(a)}{=} \lambda(\Phi_t^T S_t^2 \Phi_t + \lambda I_{\mathcal{H}})^{-1} = \lambda V_{\mathcal{D}_t}^{-1},$$

where $(a)$ follows from 4. Now given a filtration $\mathcal{G}_t$ such that $E_{1,t}$ is true, we have $I_{\mathcal{H}} - P_t \preceq \frac{\lambda}{1-\varepsilon}V_t^{-1}$, and hence $\|\Phi_t(I_{\mathcal{H}} - P_t)\|_{\mathcal{H}}^2 = \lambda_{\max}\left(\Phi_t(I_{\mathcal{H}} - P_t)\Phi_t^T\right) \leq \frac{\lambda}{1-\varepsilon}\lambda_{\max}\left(\Phi_t(\Phi_t^T\Phi_t + \lambda I_{\mathcal{H}})^{-1}\Phi_t^T\right) = \frac{\lambda}{1-\varepsilon}\lambda_{\max}\left(\Phi_t\Phi_t^T(\Phi_t\Phi_t^T + \lambda I_t)^{-1}\right) \leq \frac{\lambda}{1-\varepsilon}$. Therefore, given a filtration $\mathcal{G}_t$ such that $E_{1,t}$ is true,

$$|f(x) - \tilde{\alpha}_t(x)| \leq B\left(1 + \frac{1}{\sqrt{1-\varepsilon}}\right)\tilde{\sigma}_t(x). \tag{25}$$

Now, we have $\tilde{\mu}_t(x) = \tilde{\varphi}_t(x)^T \tilde{\theta}_t$ and $\tilde{\alpha}_t(x) = \tilde{\varphi}_t(x)^T \tilde{V}_t^{-1}\tilde{\Phi}_t^T f_t$. Also observe that $\lambda \|\tilde{\varphi}_t(x)\|_{\tilde{V}_t^{-1}}^2 = \tilde{\sigma}_t^2(x) + \tilde{\varphi}_t(x)^T\tilde{\varphi}_t(x) - k(x,x) = \tilde{\sigma}_t^2(x) - \langle \varphi(x), (I_{\mathcal{H}} - P_t)\varphi(x)\rangle_{\mathcal{H}} \leq \tilde{\sigma}_t^2(x)$, since by definition $P_t \preceq I_{\mathcal{H}}$. Then, by Cauchy-Schwartz inequality

$$|\tilde{\alpha}_t(x) - \tilde{\mu}_t(x)| \leq \left\|\tilde{V}_t^{-1}\tilde{\Phi}_t^T f_t - \tilde{\theta}_t\right\|_{\tilde{V}_t}\|\tilde{\varphi}(x)\|_{\tilde{V}_t^{-1}} \leq \lambda^{-1/2}\left\|\tilde{V}_t^{-1}\tilde{\Phi}_t^T f_t - \tilde{\theta}_t\right\|_{\tilde{V}_t}\tilde{\sigma}_t(x).$$

Now, Lemma 13 implies that for any $\delta \in (0,1]$, with probability at least $1 - \delta$, uniformly over all $t \in [T]$ and $x \in \mathcal{X}$,

$$|\tilde{\alpha}_t(x) - \tilde{\mu}_t(x)| \leq 4\sqrt{m_t/\lambda}\, v^{\frac{1}{1+\alpha}}\,(\ln(2m_t T/\delta))^{\frac{\alpha}{1+\alpha}}\, t^{\frac{1-\alpha}{2(1+\alpha)}}\tilde{\sigma}_t(x). \tag{26}$$

By triangle inequality,

$$|f(x) - \tilde{\mu}_t(x)| \leq |f(x) - \tilde{\alpha}_t(x)| + |\tilde{\alpha}_t(x) - \tilde{\mu}_t(x)|.$$

Now, combining 25 and 26, for any $\delta \in (0,1]$ and given a filtration $(\mathcal{G}_t)_{t\geq 1}$ such that $E_{1,t}$ is true for all $t \in [T]$, we have, with probability at least $1 - \delta$, uniformly over all $t \in [T]$ and $x \in \mathcal{X}$,

$$|f(x) - \tilde{\mu}_t(x)| \leq \left(B\left(1 + \frac{1}{\sqrt{1-\varepsilon}}\right) + 4\sqrt{m_t/\lambda}\, v^{\frac{1}{1+\alpha}}\,(\ln(2m_t T/\delta))^{\frac{\alpha}{1+\alpha}}\, t^{\frac{1-\alpha}{2(1+\alpha)}}\right)\tilde{\sigma}_t(x).$$

From Lemma 17, the event $E_{1,t}$ is true for all $t \in [T]$ with probability at least $1 - \delta$. Now taking an union bound, we obtain that for any $\delta \in (0,1]$, with probability at least $1 - \delta$, uniformly over all $t \in [T]$ and $x \in \mathcal{X}$,

$$|f(x) - \tilde{\mu}_t(x)| \leq \left(B\left(1 + \frac{1}{\sqrt{1-\varepsilon}}\right) + 4\sqrt{m_t/\lambda}\, v^{\frac{1}{1+\alpha}}\,(\ln(4m_t T/\delta))^{\frac{\alpha}{1+\alpha}}\, t^{\frac{1-\alpha}{2(1+\alpha)}}\right)\tilde{\sigma}_t(x).$$

Further observe that $|f(x) - \tilde{\mu}_0(x)| = |f(x)| \leq Bk^{1/2}(x,x) \leq B(1 + 1/\sqrt{1-\varepsilon})\tilde{\sigma}_0(x)$. Now, the result follows by setting $\beta_{t+1} = B\left(1 + \frac{1}{\sqrt{1-\varepsilon}}\right) + 4\sqrt{m_t/\lambda}\, v^{\frac{1}{1+\alpha}}\,(\ln(4m_t T/\delta))^{\frac{\alpha}{1+\alpha}}\, t^{\frac{1-\alpha}{2(1+\alpha)}}$ for all $t \geq 0$. ∎

Now we are ready to prove the regret bound of ATA-GP-UCB under Nyström approximation.

### D.3.4 Proof of Theorem 4

For any $\delta \in (0,1]$, we have, with probability at least $1 - \delta$, uniformly over all $t \in [T]$, the instantaneous regret

$$
\begin{aligned}
r_t &= f(x^\star) - f(x_t) \\
&\overset{(a)}{\leq} \tilde{\mu}_{t-1}(x^\star) + \beta_t \tilde{\sigma}_{t-1}(x^\star) - f(x_t) \\
&\overset{(b)}{\leq} \tilde{\mu}_{t-1}(x_t) + \beta_t \tilde{\sigma}_{t-1}(x_t) - f(x_t) \\
&\overset{(c)}{\leq} 2\beta_t \tilde{\sigma}_{t-1}(x_t)
\end{aligned}
$$

Here $(a)$ and $(c)$ follow from Lemma 20, and $(b)$ is due to the choice of ATA-GP-UCB (Algorithm 2). From Lemma 17, given a filtration $(\mathcal{G}_t)_{t\geq 1}$ such that the event $E_{2,t}$ is true for all $t \in [T]$, we have $m_t = O\left(\frac{\rho^2}{\varepsilon^2}\gamma_t \ln(T/\delta)\right)$. This, in turn, implies that

$$\beta_t = O\left(B\left(1 + \frac{1}{\sqrt{1-\varepsilon}}\right) + \frac{\rho}{\varepsilon}\sqrt{\gamma_t \ln(T/\delta)}\, v^{\frac{1}{1+\alpha}}\left(\ln\left(\frac{\gamma_t \ln(T/\delta)T}{\delta}\right)\right)^{\frac{\alpha}{1+\alpha}} t^{\frac{1-\alpha}{2(1+\alpha)}}\right).$$

Further, given a filtration $(\mathcal{G}_t)_{t\geq 1}$ such that the event $E_{1,t}$ is true for all $t \in [T]$, we have

$$\sum_{t=1}^{T} \tilde{\sigma}_{t-1}(x_t) \overset{(a)}{\leq} \rho \sum_{t=1}^{T} \sigma_{t-1}(x_t) \overset{(b)}{\leq} \rho\sqrt{T\sum_{t=1}^{T}\sigma_{t-1}^2(x_t)} \overset{(c)}{\leq} \rho\sqrt{2(1+\lambda)T\gamma_T} = O\left(\rho\sqrt{T\gamma_T}\right).$$

Here $(a)$ follows from Lemma 16, $(b)$ follows from Cauchy-Schwartz inequality, and $(c)$ follows from Lemma 6. Now from Lemma 17, with probability at least $1 - \delta$, both $E_{1,t}$ and $E_{2,t}$ are true for all $t \in [T]$. Hence, by virtue of an union bound, we obtain that for any $\delta \in (0,1]$, with probability at least $1 - \delta$, the cumulative regret of ATA-GP-UCB under Nyström appproximation after $T$ rounds is

$$R_T = O\left(\rho B\left(1 + \frac{1}{\sqrt{1-\varepsilon}}\right)\sqrt{T\gamma_T} + \frac{\rho^2}{\varepsilon}\, v^{\frac{1}{1+\alpha}}\left(\ln\left(\frac{\gamma_T \ln(T/\delta)T}{\delta}\right)\right)^{\frac{\alpha}{1+\alpha}}\sqrt{\ln(T/\delta)}\gamma_T T^{\frac{1}{1+\alpha}}\right).$$

## E   Addendum to experiments

Compared to this paper's setting, Bubeck et al. [8] makes weaker assumptions (i.e., no regularity structure on arms' rewards) and shows a more general but weaker regret bound (especially if the number of arms is very large) which is not surprising – more structure allows for lower regret. Our results show how smoothness in the arms' rewards (which is common in practice) can be exploited to achieve better regret. Numerical comparisons of the Robust-UCB algorithm (with truncated mean estimator) of Bubeck et al. [8] with our algorithms on the lightsensor data indicate that ATA-GP-UCB-Nyström performs much better than Robust-UCB, suggesting that it is indeed able

to capture the smoothness structure present in the data. Theoretically if there are only $K$ arms, the cumulative regret of ATA-GP-UCB will be better than that of Robust-UCB as long as $\gamma_T \leq K^{\frac{\alpha}{1+\alpha}}$. This holds if $K^{\frac{\alpha}{1+\alpha}} \geq (\ln T)^d$ for SE kernel and if $K^{\frac{\alpha}{1+\alpha}} \geq T^{\frac{1}{1+\nu}}$ for Matérn kernel (on $\mathbb{R}$). This is typically true if $K$ is large and in fact, for a continuous set of arms the analysis of Robust-UCB yields a trivial regret upper bound of infinity. This introduces additional challenges that require a different set of ideas and is quite representative of real world problems, e.g., hyperparameter tuning in ML.