[Reviews · NeurIPS 2019]

Reviewer 1



This work considers a challenging and important setting of heavy-tailed rewards and combines a couple of existing techniques and ideas in a novel way: regret lower bound proof (Scarlett et al.), adaptive truncation in feature space (Shao et al.) and kernel approximation techniques (Mutny et al., Calandriello et al.). While the same problem has been considered in the k-armed bandit setting (and linear bandits) before, the extension to the kernelized bandits setting is new and introduces additional challenges that require a different set of ideas. The paper is well-organized and clearly written. The exposition is clear and a great number of remarks help to better understand the obtained results in the context of previous works. The paper structure makes a lot of sense: a suboptimal algorithm that is easy to understand -> regret lower bound -> main algorithm and almost optimal regret upper bound result. The related work is also adequately cited in my opinion. Apart from the previously mentioned works, the related works on kernel ridge regression, GP-UCB, relevant stochastic bandit literature, and other BO works are well covered. Unlike some previous works in BO that used kernel approximation schemes to reduce the computational cost of inverting kernel matrices (in, e.g., works on high-dimensional BO), this work uses kernel approximation schemes to obtain almost optimal regret bounds. This is done by adaptive truncation of features, while at the same time keeping the approximation error low (the latter was also done in, e.g., Calandriello et al.). The connection between these two ideas is novel to my knowledge. I did not check the proofs in the appendix, but the main presented results seem convincing and the overall analysis certainly looks non-trivial. The novelty in the lower bound proof is the construction of the reward distribution which is specific to this particular setting while the rest of the analysis closely follows the one from Scarlett et al. When it comes to TGP-UCB, I also agree that blowing up the confidence bounds by a multiplicative factor of b_t in data space provides valid confidence intervals, but I am wondering if switching to the feature space (+ kernel approximation) is indeed necessary to obtain the optimal regret (and confidence intervals) bounds, and if yes, why precisely would that be the case? In ATA-GP-UCB, the ridge-leverage-scores-based weighting of different directions is used together with the adaptive truncation to compute approximate rewards. However, TGP-UCB performs truncation only, which leads to suboptimal regret bounds (e.g., in the case of SE kernel) even when reduced to the sub-Gaussian environment. This makes me wonder if some other raw observation truncation ideas can lead to at least a better (if not optimal) regret bound than the one obtained for TGP-UCB. Minor comments: Unlike the exploration (confidence) parameter in TGP-UCB, ATA-GP-UCB requires the knowledge of the time horizon T. Unlike the sub-Gaussian environment, here, one needs to estimate two parameters alpha and v. Perhaps, it is worth testing empirically (on synthetic data) for the robustness of the obtained confidence sets (e.g., from Lemma 1) with respect to the misestimation of these two parameters. Finally, this work also complements the standard confidence bound results (from Srinivas et al., Abbasi et al., Chowdhury et al., Durand et al, etc.) with the new ones in the case of heavy-tailed rewards. Both the result and the technique used to obtain the tighter confidence sets that depend on the approximation error (similarly to Mutny et al. and Calandriello et al.) can be of interest to the BO/GP bandit community. ---------------------------- Edit after author rebuttal: After reading the rebuttal and other reviews I remain confident that this is high-quality work and should be accepted.

Reviewer 2



The paper focuses on the bayesian optimisation with heavy tailed noise. The goal is to maximize a function f over an action space X in a bandit setting. When chosing the action x, the player observes f(x) + some heavy tailed noise. f has some regularity: it is bounded in some RKHS norm. The authors first present a simple algorithm which is far from being optimal. They give lower bounds for specific kernels: SE and Matern. They then present an algorithm with two different embedding strategies, their regret bounds and compare them empirically. The used methods are very interesting and the black box optimization is a significant problem. The theoretical results are nice. However, I have the general feeling that the authors tend to oversell their results: the algorithm does not seem to be adapted to continuous set of arms and does not seem to have a nice complexity in practice as well. I am not yet convinced of the improvement brought by this work, especially compared to the "bandits with heavy tail" paper. For these reasons, I give a weak reject score to the paper. ------------------------------------------------- Major comments: 1. You first consider the problem with a possibly continuous set of arms. However, the computation of your algorithm scales linearly with the cardinal of this set, meaning that your algorithm does only work for a finite set of arms. Besides losing a lot of interest in the original problem, this is the kind of oversold results I mentioned. 2. Let us now consider only finite set of actions. The bandits setting seems to be a more general formulation than your problem then: no regularity of f is assumed. In this case, you need to compare your results (theoretically and empirically) with what is obtained by the bandits setting. More generally, it is hard to place this paper with the related literature. I am especially eager to know the performance of the "bandits with heavy tail" algorithm in the experiments with real data. 3. The complexity of the algorithm (see page 7) is very large (besides being linear in the cardinality of X) and this would make your algorithm not very practical. Moreover, you claim some complexity if gamma << T. First, what does this mean ? Second, gamma can be a power of T (eg in the Matern kernel according to line 127), so this is an overstatement to me. Also, why isn't the per step space complexity of Nystrom embedding scaling with t (but only with m_t) ? We indeed need to store all x_t' for t'

Reviewer 3



This paper addresses Bayesian optimization for an objective function belonging to the RKHS induced by some kernel with heavy-tailed non-subGaussian noise. The first algorithm simply applies GP-UCB with truncated reward, and achieves sublinear regret for general kernels though it is not optimal. The second algorithm uses approximation of a kernel by features with appropriately chosen dimension and makes truncation depending on each feature, which matches the regret lower bound for some kernels. The formulation of the problem is well-motivated and the regret lower and upper bounds are highly non-trivial. In addition, the kernel approximation in the framework of the (theoretical analysis of) BO is quite interesting and seems to be of independent interest for general BO framework (not only with heavy-tailed noise). The following are few minor concerns for the paper. - The truncation strategy in this paper maps reward (or its counterpart for the ATA-GP-UCB) larger than b_t to 0, not b_t. As far as I scan the appendix it seems to make no problem, but it is somewhat not intuitive since the truncated reward changes non-continuously. So it would be beneficial if there is some explanation (from theoretical and/or experimental viewpoints). - It is unclear on the dimension of the QFF embedding. As discussed around Lemma 1, m_t should not be too large. Nevertheless, Theorem 3 only specifies a lower bound of m for the regret bound. It seems from the appendix that the order of m must be equal to (and no larger than) the specified value in Theorem 3. - Regarding the dimension of the kernel approximation, it would be interesting to compare the empirical results for different choice of m (or q) to evaluate how the requirement (m should not be too large) is essential rather than for the theoretical analysis. - In Line 338, I don't understand what "we simulate rewards as y(x) = f(x) + eta, where eta takes values in {-10,10}, uniformly, for any single random point in X, and is zero everywhere else" means. Isn't a single point in a real space is picked with probability zero? - Remark 7: "Theorem 3 and 4" should be "Theorems 3 and 4". ------------ The feedback is convincing to me and please explicitly write them in the final version since they are unclear from the current manuscript.

[Author Response · NeurIPS 2019]

We thank all the reviewers for their careful reviews and insightful feedbacks.

---

*Reviewer 1:* **1.** Analysis of TGP-UCB depends on a Hoeffding type concentration bound for self-normalized processes and analysis of ATA-GP-UCB depends on a Bernstein type concentration bound in each direction. Thus, if we can derive a Bernstein type bound for self-normalized processes which depends on the $(1 + \alpha)$-th moments of rewards, then we may not have to switch to the feature space to get optimal regret. **2.** Instead of truncating raw observations to build a robust estimator of $f$, one can use a median of means type estimator to obtain a slightly better bound, though not optimal, for TGP-UCB. **3.** ATA-GP-UCB requires knowledge of the horizon $T$, but one can use a standard doubling trick to get around this. **4.** It is indeed an interesting direction to explore when $\alpha$ and $v$ are misspecified. In practice, as suggested by our real data experiments, one can estimate $\alpha$ and $v$ from data and use those in our algorithms.

---

*Reviewer 2:* **1. Computational complexity for continuum arm sets:** One of our main aims is to quantify the achievable statistical efficiency of nonparametric algorithms for optimization under heavy-tailed noise from a theoretical standpoint. This is the reason why we do not delve in detail into the specifics of how the function $g_t(x) := \tilde{\mu}_{t-1}(x) + \beta_t \tilde{\sigma}_{t-1}(x)$ over all $x \in \mathcal{X} \subset \mathbb{R}^d$ is optimized. This practical issue also arises in other well-known bandit optimization settings such as finite dimensional linear bandits – the celebrated LinUCB or OFUL algorithms do not address how to solve the UCB optimization problem over a continuum set. In this regard, we mention that it is well known in the BO literature that one can approximately maximize $g_t$ by grid search / Branch and Bound methods such as DIRECT (Brochu et al. (2010)). In fact $g_t$ can be maximized within $O(\epsilon)$ accuracy by making $O(\epsilon^{-d})$ calls to it, yielding a per-step time complexity of $O(m_t^2(t + \epsilon^{-d}))$. Another viewpoint for considering a continuum arm set is that it serves to model the case of a large, finite set of arms which share regularity structure with respect to their rewards, enforced through a kernel function, and makes the analysis of the algorithm cleaner due to results in Gaussian process theory. **2. Versus the "bandits with heavy tail" paper:** Compared to this paper's setting, "bandits with heavy tail" indeed makes weaker assumptions (i.e., no regularity structure on arms' rewards) and shows a more general but weaker regret bound (especially if the number of arms is very large) which is not surprising – more structure allows for lower regret. Our results show how smoothness in the arms' rewards (which is common in practice) can be exploited to achieve better regret. Numerical comparisons of the Robust-UCB algorithm (with truncated mean estimator) of "bandits with heavy tail" paper with our algorithms on real datasets (figure is given for lightsensor data) indicate that ATA-GP-UCB-Nyström performs much better than Robust-UCB, suggesting that it is indeed able

to capture the smoothness structure present in the data. Theoretically if there are only $K$ arms, the cumulative regret of ATA-GP-UCB will be better than that of Robust-UCB as long as $\gamma_T \leq K^{\frac{\alpha}{1+\alpha}}$. This holds if $K^{\frac{\alpha}{1+\alpha}} \geq (\ln T)^d$ for SE kernel and if $K^{\frac{\alpha}{1+\alpha}} \geq T^{\frac{1}{1+\nu}}$ for Matérn kernel (on $\mathbb{R}$). This is typically true if $K$ is large and in fact, for a continuous set of arms the analysis of Robust-UCB yields a trivial regret upper bound of infinity. This introduces additional challenges that require a different set of ideas and is quite representative of real world problems, e.g., hyperparameter tuning in ML. **3. More details on time and space complexity:** For SE kernel $\gamma_t$ is poly-logarithmic in $t$ (i.e., $\gamma_t << t$), and since $m_t = \tilde{O}(\gamma_t)$, per step time complexity, in practice, for continuous $\mathcal{X}$ is $\tilde{O}(t + \epsilon^{-d})$. For Matérn kernel, the complexity is $\tilde{O}(t^p(t + \epsilon^{-d})), 1 < p < 2$. Now for finite $\mathcal{X}$, we only need to store the number of times each arm has been played so far and thus per step space complexity does not grow linearly with $t$. For continuous $\mathcal{X}$, we indeed need to store all previously chosen arms, but this linear dependence on $t$ is subsumed by the larger $|\mathcal{X}|$ term. Thus in practice, the space needed is $O(t + m_t(m_t + \epsilon^{-d})) = O(m_t(m_t + \epsilon^{-d}))$ for small enough $\epsilon$. So for continuous $\mathcal{X}$ and for practical purposes, time and space complexities can be obtained by replacing $|\mathcal{X}|$ with $\epsilon^{-d}$ in those given in the paper. **6, 8, 9:** These are typos. **7:** $\mathcal{X}_t$ is the set $\{x_1, \ldots, x_t\}$. **10.** Setting $\delta = 1/T$, we can achieve expected cumulative regret of same order (upto some constant factor). **11:** This holds, for example, Matérn kernel on $\mathbb{R}^2$ with $\nu = 3.5$ when variance of the rewards is finite. **13:** True, our algorithm is not optimal for Matérn kernel as mentioned in Remark 7. **14:** We meant to say that existing BO algorithms like GP-UCB fails under heavy-tailed noise (figure 1(f)).
*Reference:* Brochu, Eric, Cora, Vlad M., and de Freitas, Nando. A Tutorial on Bayesian Optimization of Expensive Cost Functions, with Application to Active User Modeling and Hierarchical Reinforcement Learning. *CoRR*, 2010.

---

*Reviewer 3:* **1.** Setting $y_t = 0$ if $|y_t| > b_t$ and blowing up the confidence width of GP-UCB by $b_t$, together help us to construct a robust estimate of $f$ and a "good" confidence set around this estimate which contains $f$ (figure 1(f)). Instead of truncating, one can also use a median of means type estimator in the feature space and obtain optimal regret. **2.** The result will hold as long as $\bar{m} > 1/l^2$ and $\bar{m} = \Theta(2 \log_{4/e}(T^3))$. **3.** We compared for different choices of $\bar{m}$ and as long as $\bar{m}$ satisfies the above constraints, we found that increasing $\bar{m}$ improves the performance to some extent. **4.** We discretized the set $[0, 1]$ into 100 evenly spaced points and generated a random sample uniformly from those 100 points.

[Meta-Review · NeurIPS 2019]

This is a strong submission and the author did an excellent job with the rebuttal. Please make sure that this makes its way to the camera ready version.